# On A Mallows-type Model For (Ranked) Choices

**Yifan Feng**
Department of Analytics and Operations
NUS Business School
yifan.feng@nus.edu.sg

**Yuxuan Tang**
Institute of Operations Research and Analytics
National University of Singapore
yuxuan.tang@u.nus.edu

## Abstract

We consider a preference learning setting where every participant chooses an ordered list of $k$ most preferred items among a displayed set of candidates. (The set can be different for every participant.) We identify a distance-based ranking model for the population's preferences and their (ranked) choice behavior. The ranking model resembles the Mallows model but uses a new distance function called Reverse Major Index (RMJ). We find that despite the need to sum over all permutations, the RMJ-based ranking distribution aggregates into (ranked) choice probabilities with simple closed-form expression. We develop effective methods to estimate the model parameters and showcase their generalization power using real data, especially when there is a limited variety of display sets.

## 1 Introduction

How to aggregate the population's preferences from their (ranked) choices out of different choice sets? This question is of interest to many communities, such as economics, business, and computer science. A concrete setting is a platform that wishes to learn customer preferences over a universe of $n$ product prototypes. The platform is able to display different subsets of versions to different customers, who then provide feedback in the form of a top-$k$ ranked list of their most preferred items within the subset they see (hereafter referred to as "ranked choices").[1]

The population's (ranked) choice behavior could be summarized as a *(ranked) choice model* $\Pr(\pi_k|S)$, which specifies the probability that a randomly drawn participant choosing a top-$k$ list $\pi_k$ from the display set $S$. An economically rationalizable and yet very general way to model (ranked) choices is to use probabilistic *ranking models*. That is, given a probability distribution $\lambda$ over preference rankings, $\Pr(\pi_k|S)$ equals the probability that a randomly drawn participant would place $\pi_k$ as the top-$k$ positions among the items in $S$.

A popular family of ranking models is *distance-based*, which is the conceptual analog of Gaussian distribution for scalars; see [1]. A distance-based ranking model is specified by a modal (central) ranking $\pi^*$ and a dispersion parameter $q \in (0, 1)$. Given a ranking $\pi$, its probability of being sampled is proportional to $q^{d(\pi^*, \pi)}$. Here $d(\pi_1, \pi_2)$ is a *distance function* that describes the discrepancy of ranking $\pi_1$ from $\pi_2$, and different distance functions lead to different models. The most popular distance-based ranking model is the *Mallows model* ([2]), which uses the Kendall-Tau distance as its distance function. It has been studied extensively in the literature regarding topics such as sampling, estimation from sampled (partial) rankings, and learning in a Mallows mixture setting; see [3] and references therein. Due to its popularity and the fact that every distance-based ranking model only differs from the Mallows by distance function, we will also refer to a distance-based ranking model as a *Mallows-type model* interchangeably in the sequel.

---

[1]If $k = 1$, each participant is asked to choose the most preferred candidate. That effectively reduces to a (single) choice, which is a feedback structure extensively studied in the choice modeling literature.

36th Conference on Neural Information Processing Systems (NeurIPS 2022).

Mallows-type models could be used as "kernels" to "smooth out" the distribution over rankings. As such, they could help mitigate the overfitting issues of generic (i.e., nonparametric) ranking models, which are typically overparameterized. Yet, Mallows models can be highly expressive in a mixture setting: A mixture of Mallows-type models can approximate *any* probability distribution over rankings by an arbitrary precision (as $q$ tends to zero and the number of clusters tends to infinity). Therefore, the Mallows-type model family is a helpful tool to balance capturing (complex) preference heterogeneity across individuals and regularizing the ranking distribution for better out-of-sample predictions. A representative work is by Antoine et al. [4], who use the Mallows model to aggregate customer preferences from their choices (i.e., $k = 1$).

Despite the theoretical elegance, the main challenge in applying Mallows-type ranking models to (ranked) choice modeling is analytical and computational tractability. More specifically, $\Pr(\pi_k|S)$ is calculated from summation over all rankings subject to nontrivial conditions. Therefore, even if a Mallows-type ranking model has a simple structure for $\lambda$, the resulting choice probabilities $\{\Pr(\pi_k|S)\}$ can be difficult to obtain even when $k = 1$. The state-of-art results are obtained by Antoine et al. [4], who develop polynomial-time numerical algorithms to compute $\{\Pr(\pi_k|S)\}$ for $k = 1$ under the Mallows model. The *estimation* problem is even more difficult, which involves finding the central ranking $\pi^*$ and dispersion parameters $q$ that best explain the (ranked) choice data. To the best of our knowledge, no effective methods to estimate *any* Mallows-type ranking model from (ranked) choice data are known. (Perhaps the best method to date is again by Antoine et al. [4], who use a "Mallows smoothing" heuristic to conduct the estimation when $k = 1$, which we will discuss later.)

**Summary of results and contributions.** This paper identifies and studies a new distance-based (i.e., Mallows-type) ranking model. It is the same as the Mallows model except that it builds on a new distance function (i.e., smoothing kernel), which we call *reverse major index (RMJ)*. Unlike the Mallows model's Kendall-Tau distance (which weighs all pairwise disagreements equally), RMJ puts more weight on top-position deviations.

The RMJ-based ranking model is a small conceptual deviation from the Mallows model (and, in particular, enjoys the desired properties mentioned above, such as rationalizability, smoothing, and expressive power); see Section 2. However, this twist brings a significant advantage in both analytical and computational simplicity. Specifically, we solve a list of problems under the RMJ-based ranking distribution. That includes:

- *Characterizing (ranked) choice probabilities*: Given $k \geq 1$, calculating $\{\Pr(\pi_k|S)\}$;
- *Sampling*: Given $k \geq 1$, efficiently sampling a top-$k$ list $\pi_k$;
- *Parameter learning*: Estimating the central ranking $\pi^*$ and dispersion parameter $q$ from the (ranked) choice data through a maximum likelihood estimator (MLE) formulation ;
- *Learning in a mixture setting*: Assuming that there are multiple clusters of participant preferences, learning the central ranking and dispersion parameter for each cluster from choice data.

The solutions to the problems above can be implemented relatively easily. First, we are able to obtain $\{\Pr(\pi_k|S)\}$ in *simple* and *closed-form* expressions for all $k \geq 1$; see Theorems 1 and 4. This is in contrast to, say, its counterpart for the Mallows model ([4]), and we view it as our main theoretical achievement. Second, the sampling can be done in $O(nk)$ time directly; see Lemma 2. Third, the estimation problem can be reduced to a well-studied ranking-aggregation-type formulation; see (5) and Theorem 5. Many off-the-shelf tools are available. For example, it admits a polynomial-time approximation scheme (PTAS) and can be practically solved via a linear integer programming formulation. The estimation is guaranteed to recover the model parameters asymptotically under mild conditions on the coverage of display sets; see Theorem 2. This stands in contrast to Mallows Smoothing by [4], which cannot recover the Mallows central ranking even under sufficient coverage; see Theorem 3. Finally, the learning problem in a mixture setting can be solved using the standard Expectation-Maximization (EM) algorithm.

We demonstrate the practical effectiveness of our methodology on two data sets on customer preference over different types of sushi. When $k = 1$, we compare it with two representative ranking-based choice models: one based on the Mallows and the other on Plackett-Luce (which leads to the Multinomial Logit choice model). Our tools display superb generalization power, especially when there is a limited variety of display sets in the choice data. When $k > 1$, we demonstrate the robustness

of the methodology. (It is difficult to find direct comparisons for prediction power.) Specifically, as long as the underlying population keeps the same, different top-$k$ lists aggregate into the same central rankings. We also show that our methodology can handle a relatively large number of items ($n = 100$) by achieving high precision solutions ($< 2\%$ optimality gap) in a reasonable time ($< 5$ minute solving time).

**Related literature.** Our paper is most closely related to the (extensive) literature on Mallows-type ranking models and their applications to preference and choice modeling (e.g., [1, 5, 3, 6, 7, 8, 9, 4, 10] to name a few.) Besides the new distance function and tractability results explained above, the most notable difference of this paper is the *feedback structure*. In our paper, every participant chooses from an *arbitrary* display set (as opposed to pairwise or the full display set only), and their feedback is in the form of a top-$k$ ranked list (as opposed to single choices, e.g., [11, 12], or full rankings). To the best of our knowledge, the combination of both generalities makes the setting the first of its kind.[2] It is also worth mentioning that the analytical and computational simplicity of our model makes it a convenient building block for subsequent optimization problems (e.g., new product introduction). It is an advantage that many common methods (e.g., Monte-Carlo algorithms) do not have.

Also related is the literature on *learning to rank* (e.g., [14, 15]). We wish to stress that we do not merely fit a ranking from the data but also the uncertainty quantification of the estimate (and the whole choice model $\{\Pr(\pi_k|S)\}$). Finally, our paper is also closely related to that of [13]. They study a choice model, called the "ordinal attraction model" (OAM), that emerges from an active learning problem of consumer preferences. We rationalize the (surprisingly simple) OAM by showing that it is equivalent to the aggregated choice probabilities $\{\Pr(\pi_k|S)\}$ from the RMJ-based ranking model when $k = 1$. We view it as a nontrivial observation in its own right.

## 2   A Ranking Model Based on Reverse Major Index

**Preliminary.** We consider a universe of $n$ candidates (hereafter referred to as "items"), represented by $[n] = \{1, 2, \ldots, n\}$. We assume that every participant has a strict preference over these items, represented by a ranking (permutation). We use a bijection $\pi : [n] \to [n]$ to represent a ranking where $\pi(i)$ is the $i$th most preferred item. We sometimes find the notation $\sigma := \pi^{-1}$ helpful, where $\sigma(x)$ is the position of item $x$ in the ranking $\sigma$. We also use $x \succ_\pi y$ if item $x$ is preferred to item $y$ under $\pi$, i.e., $\pi^{-1}(x) < \pi^{-1}(y)$. For example, $\pi = (3, 1, 2)$ means $3 \succ_\pi 1 \succ_\pi 2$ and it corresponds to a "$\sigma$" notation of $\sigma = (2, 3, 1)$. Finally, we use $e$ to represent the identity ranking, $\Sigma$ for the set of all rankings over $n$ items, $\pi_k (k \geq 1)$ for a top-$k$ list, and $\Sigma_k$ for the set of all top-$k$ rankings.

**Mallows-type ranking models.** Given the distance function $d(\cdot, \cdot)$, the probability mass function of the ranking for a Mallows-type model can be written as

$$\lambda(\pi) = \frac{q^{d(\pi^*, \pi)}}{\sum_{\tilde{\pi}} q^{d(\pi^*, \tilde{\pi})}},$$

where $\pi^*$ is the central ranking, $q$ is the dispersion parameter, and $\sum_{\tilde{\pi}} q^{d(\pi^*, \tilde{\pi})}$ is the normalization constant. Intuitively, the Mallows-type model defines a population of participants whose preferences are "similar" as they are centered around a common ranking, where the probability for deviations thereof decreases exponentially.

Different distance functions correspond to different models. A common requirement for a valid distance function is that it is invariant to "relabeling." Formally, that means $d(\cdot, \cdot)$ is left-invariant under the ranking composition, i.e., $d(\pi_1, \pi_2) = d(\pi\pi_1, \pi\pi_2)$ for every $\pi, \pi_1, \pi_2$.[3] This invariance property enables to make the following conventions without loss of generality: First, we assume that the items have been properly relabeled so that the central ranking $\pi^* = e$ (unless otherwise specified). Second, we may use $d(\pi)$ as shorthand notation for $d(e, \pi)$, which fully represents a distance function with the knowledge that $d(\pi_1, \pi_2) = d(e, \pi_1^{-1}\pi_2)$.

---

[2]This setting is also practically relevant. For example, the platform may have capacity constraints on displaying how many items. The platform may also have an incentive to judiciously select the display sets to make feedback collection more efficient; see [13].

[3]Equivalently, one could use the "$\sigma$" notation and write the distance function as $\tilde{d}(\sigma_1, \sigma_2) := d(\pi_2, \pi_1)$ and $\tilde{d}$ will be right-invariant under ranking composition.

**The Mallows (Kendall's Tau distance based) ranking model.** Commonly studied distance functions include the Spearman's rank correlation, Spearman's Footrule, and the Kendall's Tau distance ([1]). Among those three, Kendall's Tau corresponds to the Mallows model and is defined as

$$d_K(\pi) = \sum_{i=1}^{n-1} \sum_{j=i+1}^{n} \mathbb{I}\{\pi(i) > \pi(j)\}. \tag{1}$$

It measures a ranking's total number of pairwise disagreements (with the identity ranking $e$). For example, consider the ranking $\pi = (4, 2, 1, 3)$. There are four pairwise disagreements: $\{(4, 2), (4, 1), (4, 3), (2, 1)\}$. Therefore, $d_K((4, 2, 1, 3)) = 4$.

As an appealing property, Kendall's tau distance leads to a tractable expression for $\lambda$. Other common distances do not have this since the normalization constant involves summing over $n!$ rankings. The constant under the Mallows model can be expressed as (see [16]):

$$\sum_{\pi} q^{d_K(\pi)} = \psi(n, q) := \prod_{i=1}^{n} \frac{1-q^i}{1-q} = \prod_{i=1}^{n} \left(1 + q + \ldots + q^{(i-1)}\right). \tag{2}$$

**The RMJ-based ranking model.** In this paper, we identify a new distance function, which we call *reverse major index (RMJ)*. It is defined as

$$d_R(\pi) := \textit{Reverse Major Index}(\pi) = \sum_{i=1}^{n-1} \mathbb{I}\{\pi(i) > \pi(i+1)\} \cdot (n - i) \tag{3}$$

Compared to Kendall's Tau in (1), RMJ focuses on *adjacent* disagreements and puts more weight on top-position disagreements. For example, consider the ranking $\pi = (4, 2, 1, 3)$ again. The only adjacent disagreements are $\{(4, 2), (2, 1)\}$. Therefore, after including the weights in (3), we have $d_R((4, 2, 1, 3)) = 3 + 2 = 5$. The name of RMJ is inspired by the *major index* from the combinatorics literature ([17]), which is defined as $d_M(\pi) = \sum_{i=1}^{n-1} \mathbb{I}\{\pi(i) > \pi(i+1)\} \cdot i$.

**Discussion: Kendall's Tau vs. RMJ.** Both metrics are conceptually similar: they measure a ranking's deviation from the identity. They coincide in many intuitive cases. For example, in the most extreme ones we have $d_K(e) = d_R(e) = 0$ and $d_K((n, \ldots, 1)) = d_R((n, \ldots, 1)) = n(n-1)/2$. Therefore, both can be used as "kernels" to smooth out the distribution over rankings. However, they emphasize the deviation in a subtly different way. Therefore, we can also find subtle rankings where their values are different. For example, $d_K((4, 2, 1, 3)) = 4$ but $d_R((4, 2, 1, 3)) = 5$.

We believe that both Kendall's Tau and RMJ are reasonable measures of ranking discrepancy, and we find it difficult to tell which kernel is necessarily "better" from a theoretical/axiomatic approach. For example, [1] specifies a set of properties that a reasonable distance function should satisfy:

1. The distance $d(\pi, \tilde{\pi}) \geq 0$, and the equality holds if and only if $\pi$ and $\tilde{\pi}$ are the same ranking;

2. The distance function $d(\cdot, \cdot)$ is invariant to relabeling. (See earlier discussion in this section, especially footnote 3.)

In this sense, both Kendall's Tau and RMJ satisfy the basic axioms for ranking distances. Meanwhile, [18] specifies a larger set of axioms so that Kendall's Tau is the unique distance function satisfying all axioms. (For example, it can be verified that Kendall's Tau is a symmetric measure while RMJ is not.) However, it could also be argued that when it comes to human beings' preferences, top-position deviations matters more than bottom-position ones. If one makes that into a axiom, it will be satisfied by RMJ but not Kendall's Tau. In this regard, there is not a measure that satisfies "all possible" axioms.

Despite the aforementioned difficulties, we will show later that the RMJ produces a more tangible ranking model for ranked choices. For example, the RMJ-based ranking model leads to simple and estimatable (ranked) choice probabilities without compromising the desirable properties of Mallows, such as rationalizability and flexibility for a mixture setting. Moreover, regarding its ability to describe preference distributions that may occur in practice, we will demonstrate its descriptive and predictive power in a case study with real preference data. Therefore, we believe that RMJ produces a promising tool that is more tailed to the application of learning population preferences from (ranked) choice data.

# 3 Analysis of the (Ranked) Choice Model $\{\Pr(\pi_k|S)\}$

In this section, we will characterize the (ranked) choice model $\{\Pr(\pi_k|S)\}$ aggregated from the RMJ-based ranking model, formally defined as

$$\Pr(\pi_k|S) = \sum_{\tilde{\pi}} \lambda(\tilde{\pi})\mathbf{I}(\pi_k, \tilde{\pi}, S),$$

where $\mathbf{I}(\pi_k, \tilde{\pi}, S)$ means the top-$k$ list $\pi_k$ is compatible with the ranking $\tilde{\pi}$ in the set $S$. That is, $\pi_k(i) \in S$ and $\pi_k(i) \succ_{\tilde{\pi}} x$ for all position $i \in \{1, \ldots, k\}$ and item $x \in S \setminus \{\pi_k(1), \ldots, \pi_k(i)\}$. Note that the summation is over rankings with nontrivial conditions. Therefore it is unclear *in priori* whether *any* distance-based ranking model can aggregate into a tractable $\Pr(\pi_k|S)$ (even for $k = 1$). We will also discuss how to estimate the ranked choice model parameters from data. In the sequel, we will write $d(\cdot) = d_R(\cdot)$ since RMJ is the distance function of interest.

## 3.1 The $k = 1$ case

When $k = 1$, a top-$k$ list model reduces to a choice model, which connects to a richer literature (e.g., [4]). Therefore, it is worth a separate discussion, which also helps build intuition for the $k > 1$ case.

**Choice probabilities.** Our main result for $k = 1$ is summarized below, which characterizes the choice probabilities under the RMJ-based ranking model.

**Theorem 1** *Let a display set $S = \{x_1, x_2, \ldots, x_M\}$ be such that $x_1 < x_2 < \cdots < x_M$. Under the RMJ-based ranking model*

$$Pr(\{x_i|S\}) = \frac{q^{i-1}}{1 + q + \ldots + q^{M-1}}. \tag{4}$$

In other words, for every display set, all the items within the display set are re-ranked so that their choice probabilities decay exponentially fast according to their *relative ranking* within the display set. Noticeably, the choice probabilities in (4) are (much) simpler than that induced by the Mallows model, which even needs a Fast Fourier Transform to evaluate in $O(n^2 \log n)$ time ([4]). Also, (4) rediscovers the "Ordinal Attraction Model" (OAM) in [13], which could also be viewed as a "multiwise" generalization of pairwise noisy comparison models (e.g., [19, 20]). OAM gets its name because the "attractiveness" (i.e., choice probability) of an item within a display set $S$ only depends on its *relative* position in $S$ and therefore is only a function of the "ordinal" information. While OAM emerges from an active learning problem of consumer preferences, we "rationalize" it by showing that it can be aggregated from the RMJ-based ranking model, which we believe is a nontrivial observation in its own right. In the sequel, we will follow its convention and refer to the choice model defined in Theorem 1 as OAM.

**Proof outline and key intermediate results for Theorem 1.** Let us introduce a few notations for top-$k$ (sub-) rankings. Given a top-$k$ ranking $\pi_k \in \Sigma_k$, we could define RMJ for $\pi_k$ by truncating the index at position $k$, i.e., $d(\pi_k) = \sum_{i=1}^{k-1} \mathbb{I}\{\pi_k(i) > \pi_k(i+1)\} \cdot (n - i)$. In addition, let $R(\pi_k) := \{\pi_k(i) : i = 1, \ldots, k\}$ be the set of top-$k$ items and $R^c(\pi_k) := [n] \setminus R(\pi_k)$ its complement. Finally, let $L(\pi_k) := |\{x \in R^c(\pi_k) : x < \pi_k(k)\}|$ be the number of items that are (i) not included in $\pi_k$ and (ii) having smaller indices than (i.e., preferred under $\pi^* = e$) item $\pi_k(k)$. For example, suppose $n = 7$ and $\pi_4 = (7, 4, 6, 2)$. Then $d(\pi_4) = 6 + 4 = 10, R(\pi_4) = \{2, 4, 6, 7\}, R^c(\pi_4) = \{1, 3, 5\}, \pi_4(4) = 2$, and $L(\pi_4) = |\{1\}| = 1$.

Given two subrankings $\pi_k \in \Sigma_k$ and $\pi_{k'} \in \Sigma_{k'}$ with $k \leq k'$, we write $\pi_k \subseteq \pi_{k'}$ if they are compatible, i.e., $\pi_k(i) = \pi_{k'}(i)$ for all $i = 1, \ldots, k$. Our first result concerns extending the domain of $\lambda$ to top-$k$ rankings, formally defined as $\lambda(\pi_k) := \sum_{\tilde{\pi}} \lambda(\tilde{\pi})\mathbb{I}\{\pi_k \subseteq \tilde{\pi}\}$.

**Lemma 1 (Probability distribution of top-$k$ rankings)** $\lambda(\pi_k) = q^{d(\pi_k)+L(\pi_k)} \cdot \frac{\psi(n-k,q)}{\psi(n,q)}$.

The significance of the result above is that we could think of a ranking $\pi$ as a stochastic process on a "tree" with depth $n$ and $n!$ leaves. While $\{\lambda(\pi) : \pi \in \Sigma\}$ describe the probability distribution over the leaves, for every $k$, $\{\lambda(\pi_k) : \pi_k \in \Sigma_k\}$ describe the probability distribution over the nodes at level $k$.

As a consequence of Lemma 1, we can write out how to randomly sample a ranking $\pi$ under the RMJ-based ranking model from the top to the bottom position. Given a top-$k$ ranking $\pi_k$ and an item $z$, we write $\pi_k \oplus z$ as the concatenation of $\pi_k$ and item $z$. For example, suppose $\pi_3 = (5, 2, 4)$, then $\pi_3 \oplus 3 = (5, 2, 4, 3)$.

**Lemma 2 (Random ranking generation)** *Given a top-k ranking $\pi_k$ such that $\pi_k(k) = z$, the conditional probability for the $(k+1)$-positioned item is*

$$Pr(\pi_{k+1} = \pi_k \oplus y | \pi_k) := \frac{\lambda(\pi_k \oplus y)}{\lambda(\pi_k)} = \frac{q^{h(y|z)-1}}{1+q+\cdots+q^{n-k-1}},$$

*where* $h(y|z) = \begin{cases} \sum_{x \in R^c(\pi_k)} \mathbb{I}\{z < x \le y\} & \text{if } y > z \\ n - k - \sum_{x \in R^c(\pi_k)} \mathbb{I}\{y < x < z\} & \text{if } y < z \end{cases}.$

Finally, Lemmas 1 and 2 lead to the following result based on an induction argument.

**Lemma 3** *Let a display set $S = \{x_1, x_2, \ldots, x_M\}$ be such that $x_1 < x_2 < \cdots < x_M$ and a top-k ranking $\pi_k$ be such that $\pi_k = \pi_{k-1} \oplus z$ and $R(\pi_{k-1}) \cap S = \emptyset$. Then conditional on a participant's top-k preference list is $\pi_k$, the probability that (s)he will choose item $x_i$ out of display set $S$ is*

$$Pr(\{x_i | S\} | \pi_k) := \frac{\sum_{\tilde{\pi}} \lambda(\tilde{\pi}) \cdot \mathbf{I}(x_i, \tilde{\pi}, S) \cdot \mathbb{I}\{\pi_k \subseteq \tilde{\pi}\}}{\lambda(\pi_k)} = \begin{cases} \frac{q^{M-p(z|S)+i-1}}{1+q+\ldots+q^{M-1}} & \text{if } z > x_i \text{ and } z \notin S, \\ \frac{q^{i-p(z|S)-1}}{1+q+\ldots+q^{M-1}} & \text{if } z < x_i \text{ and } z \notin S, \\ 1 & \text{if } z = x_i, \\ 0 & \text{if } z \in S \setminus \{x_i\}. \end{cases}$$

*where* $p(z|S) := \sum_{x \in S} \mathbb{I}\{x < z\}$.

Note that Lemma 3 could be viewed as a generalization of Theorem 1 by setting $\pi_k = \emptyset$ (which corresponds to $z = 0$).

**Parameter learning from choice data.** In the parameter learning problem, we are endowed with choice data $H_T = (S_1, x_1, \ldots, S_T, x_T)$, where $S_t$ is the display set shown to the $t$th participant and $x_t$ is his/her choice. Following [13], the maximum likelihood estimator (MLE) for the central ranking $\pi^*$ can be obtained from the following *choice aggregation* problem

$$\hat{\pi} \in \text{argmin}_\pi \sum_{t=1}^{T} \sum_{x \in S_t} \mathbb{I}\{x \succ_\pi x_t\}.$$

It has a further linear integer programming formulation. Let $w_{ij} := \sum_{t=1}^{T} \mathbb{I}\{\{i, j\} \subseteq S_t \text{ and } x_t = i\}$ be the number of times that item $i$ and item $j$ are displayed together and item $i$ is chosen. Intuitively, a positive $w_{ij} - w_{ji}$ is an indication that item $i$ should be preferred to item $j$. Invoking Proposition 3 in [13], the integer programming could be written as follows:

$$
\begin{aligned}
\hat{x} \in \text{argmin}_x \quad & \sum_{(i,j):i \ne j} w_{ij} x_{ji} \\
\text{s.t.} \quad & x_{ij} + x_{ji} = 1 && \forall 1 \le i, j \le n, i \ne j \\
& x_{ij} + x_{jr} + x_{ri} \le 2 && \forall 1 \le i, j, r \le n, i \ne j \ne r \\
& x_{ij} \in \{0, 1\} && \forall 1 \le i, j \le n
\end{aligned}
\tag{5}
$$

In the formulation above, the solution $\hat{x}$ is such that $\hat{x}_{ij} = 1$ if $i \succ_{\hat{\pi}} j$ under the MLE $\hat{\pi}$. Computationally, this integer programming is an instance of the well-studied *feedback arc set problem on tournaments*. Therefore, it admits a polynomial-time approximation scheme (PTAS). From a more practical side, the central ranking $\pi^*$ can be effectively obtained using off-the-shelf integer programming solvers and with many speeding-up heuristics; see [13] for more discussion. The MLE for dispersion parameter $q$ can be subsequently obtained from an one-dimensional (convex) optimization problem $\hat{\alpha} \in \text{argmin}_\alpha \{\alpha \sum_{t=1}^{T} \sum_{x \in S_t} \mathbb{I}\{x \succ_{\hat{\pi}} x_t\} + \sum_{t=1}^{T} \log \sum_{j=0}^{|S_t|-1} e^{-j\alpha}\}$ so that $\hat{\alpha} = -\ln \hat{q}$. It is also worth noting that the MLE framework can be easily extended to learning in a

mixture setting using the standard EM algorithm, which we refer to the supplementary materials for more details.

Let us conclude this section by providing some theoretical understanding of whether we could recover the ground truth values of $\pi^*$ and $q$ at least asymptotically. Intuitively, this should depend on the "coverage" of display sets: if only a pair $\{x, y\}$ is repeatedly, there is no hope of recovering the full ranking $\pi^*$. It turns out that the parameters can be recovered as long as every pair is "covered" by some display set.

**Theorem 2** *Consider a sequence of display sets $\{S_t\}$. Suppose every pair of items $\{i, j\} \subseteq [n]$ is displayed infinitely often. That is, $\sum_{t=1}^{T} \mathbf{I}\{\{i, j \in S_t\}\} \to \infty$ as $T \to \infty$. Then the MLE $(\hat{\pi}, \hat{q}) = (\hat{\pi}(H_T), \hat{q}(H_T))$ is an consistent estimator. That is, $(\hat{\pi}, \hat{q}) \to (\pi^*, q^*)$ almost surely as $T \to \infty$. Conversely, if there exists a pair of items $\{i, j\}$ that is only displayed finitely often, then there exists a tie-breaking rule for MLE so that $\hat{\pi} \not\to \pi^*$ with positive probability.*

We would like to mention that the coverage condition is fairly mild. For example, as long as the full display set is displayed sufficiently many times, the RMJ-implied central ranking can eventually be recovered. This stands in contrast to its counterpart in the Mallows model. Since the Mallows model does not produce simple choice probabilities, the MLE from choice data is rather difficult to obtain. Perhaps the best method to date is by [4], who uses a "Mallows smoothing" heuristic. The following result reveals that the heuristic can be unstable or fail to recover the underlying Mallows parameters even under sufficient coverage of display sets.

**Theorem 3** *Even if all display sets with sizes larger than two are displayed infinite times, the estimator from the Mallows Smoothing heuristic is not consistent.*

The intuition behind the result above is that Mallows Smoothing needs a "choice to ranking" step. That is, it needs to first find a distribution over rankings, denoted by $\hat{\lambda} \in \Delta(\Sigma)$, that aggregates into choice probabilities to match the empirical choice probabilities $\{\hat{\Pr}(x|S)\}$. Formally, it corresponds to solving the system of linear equations

$$\sum_{\tilde{\pi}} \hat{\lambda}(\tilde{\pi}) \mathbf{I}(x, \tilde{\pi}, S) = \hat{\Pr}(x|S) \tag{6}$$

for all display set $S$ in the data and $x \in S$. Since there are vastly more variables than equations, the solutions in general form a polytope rather than a singleton. More importantly, we find that there can be solutions $\hat{\lambda}_1$ and $\hat{\lambda}_2$ that aggregate to different Mallows models in the "ranking aggregation" step of the heuristic. Therefore, in general, this heuristic can produce non-unique results and thus be inconsistent.

## 3.2 The general ($k \geq 1$) case

In the general ($k \geq 1$) case, the ranked choice probabilities and the corresponding estimation remain parsimonious. Roughly speaking, the ranked choice model behaves like a generalized OAM where $\Pr(\pi_k|S)$ depends on the *relative rankings* of the items in $\pi_k$ among the set $S$. The tractability comes from the sole dependence on the *relative* ranking. That is, to obtain $\Pr(\pi_k|S)$, one could first treat $S$ as the "full display set" for a subuniverse of items, then re-rank all the items within the display set $S$, and finally just apply Lemma 1.

**Choice probabilities.** Given a display set $S$, let $d_S(\cdot), L_S(\cdot)$ be the originally defined $d(\cdot)$ and $L(\cdot)$ functions, but treating the display set $S$ as the universe. Formally, $d_S(\pi_k) := \sum_{i=1}^{k-1} \mathbb{I}\{\pi_k(i) > \pi_k(i+1)\} \cdot (|S| - i)$ and $L_S(\pi_k) := |\{x \in R^c(\pi_k) \cap S : x < \pi_k(k)\}|$. For example, suppose again that $n = 7$ and $\pi_4 = (7, 4, 6, 2)$, and let $S = \{2, 3, 4, 5, 6, 7\}$. Recall that $d(\pi_4) = 6 + 4 = 10$ and $L(\pi_4) = |\{1\}| = 1$. In comparison, $d_S(\pi_4) = 5 + 3 = 8$ and $L_S(\pi_4) = |\emptyset| = 0$.

**Theorem 4** *Given a display set $S$ and a top-$k$ ranking $\pi_k$ such that $R(\pi_k) \subseteq S$, we have*

$$Pr(\pi_k|S) = q^{d_S(\pi_k) + L_S(\pi_k)} \cdot \frac{\psi(|S| - k, q)}{\psi(|S|, q)}.$$

Note that Theorem 4 generalizes both Theorem 1 and Lemma 1 (but in different ways). It reduces to Theorem 1 by setting $k = 1$: Given $\pi_1 = (z)$ where $z \in S$, we have $d_S(\pi_1) = 0$, $L_S(\pi_1) = |\{x \in S \setminus \{z\} : x < z\}|$ the relative ranking of $z$ in display set $S$, and $\frac{\psi(|S|-k,q)}{\psi(|S|,q)} = 1 + q + \cdots + q^{|S|-1}$ the appropriate normalizing constant where $k = 1$. In addition, Theorem 4 reduces to Lemma 1 by setting $S = [n]$, as can be easily verified.

**Parameter learning from ranked choice data.** In the parameter learning problem, we are endowed with choice data $H_T = (S_1, \pi_k^1, \ldots, S_T, \pi_k^T)$, where $S_t$ is the display set shown to the $t$th participant and $\pi_k^t = (x_1^t, \ldots, x_k^t)$ are his/her top-$k$ choices ranked from the most preferred to the $k$th preferred.

It turns out that (somewhat surprisingly) in the $k \geq 1$ case, the central ranking $\pi^*$ can be estimated from the same integer programming (5) but just with a generalized re-definition of the weight parameters $\{w_{ij}\}$.

**Theorem 5** *The MLE for the central ranking $\pi^*$, $\hat{\pi}$, can be obtained from the integer program (5) with $\hat{x}_{ij} = 1$ iff $i \succ_{\hat{\pi}} j$ and a generalized definition of $\{w_{ij}\}$ below*

$$w_{ij} = \sum_{t=1}^{T} \left[ \mathbb{I}\{x_k^t = i\} \cdot \mathbb{I}\{\{i,j\} \subseteq S_t \setminus \{x_1^t, \ldots, x_{k-1}^t\}\} + \sum_{h=1}^{k-1} (|S_t| - h) \cdot \mathbb{I}\{x_h^t = i, x_{h+1}^t = j\} \right].$$

Intuitively, a positive $w_{ij} - w_{ji}$ is still an indication that item $i$ should be preferred to item $j$, but now taking into consideration that every participant's response is actually a *ranked list* rather than a single choice. Practically speaking, the $\{w_{ij}\}$ can be maintained in a relatively simple manner. Suppose a display set $S$ and a ranked choice $\pi_k$ are given. Then $w_{\pi_k(\ell), \pi_k(\ell+1)}$ will be added by $|S| - \ell$ for all $\ell = 1, \ldots, k-1$. (This captures the ranking information for items included in $\pi_k$.) In addition, $w_{\pi_k(k), j}$ will be added by 1 for all $\ell \in \{R^c(\pi_k) \cap S\}$. (This captures the ranking information between $\pi_k(k)$ and all items that are not included in $\pi_k$.) For example, suppose $n = 6$ and $k = 3$. The ranked choice data is such that $S_1 = \{1, 2, 3, 4, 5\}$ and $\pi_k^1 = (3, 1, 2)$. Then we have $w_{3,1} = 4, w_{1,2} = 3, w_{2,4} = w_{2,5} = 1$ and all the other $w_{ij} = 0$.

**Discussion: model limitation.** A few more words on the ranked choice model $\{Pr(\pi_k|S)\}$. The specific form gives it great simplicity, but as with any other parametric model, it may be vulnerable to model missspecification issues in practice. In particular, since the model is calculated from a unimodal ranking distribution, this model (in its "vanilla" form) is best suited when there is an approximate "consensus" ranking in the population. In our numerical studies, we will demonstrate how we can effectively mitigate those issues by considering learning the model parameters in a mixture setting to better capture the preference heterogeneity in the population.

## 4 Numerical Experiments

We investigate the performance of our methodology on two anonymous survey data sets regarding sushi preferences ([21]): The first one consists of 5,000 full preference rankings over 10 kinds of sushi, while the second one consists of 5,000 top-10 rankings over 100 kinds.[4] Throughout the numerical experiments, we use a workstation with dual Intel Xeon Gold 6244 CPU (3.6 GHz and 32 cores in total), no GPUs, and 754 GB of memory.

**Experiment 1: Top-1 choice.** When $k = 1$, a top-$k$ list corresponds to a choice model, which connects to a richer literature (e.g., [22, 4, 13]). We compare our prediction power with two representative ranking models: Mallows and Plackett-Luce. (The previous is distance-based. The latter is not but leads to the famous multinomial logit model (MNL) choice model.) We use the Mallows Smoothing (MS) heuristic to estimate a Mallows model and MLE to estimate a Plackett-Luce. For all three rankings models (Mallows, PL, and ours), we also use an EM algorithm to perform parameter learning in a mixture setting.

Since the estimation and generalization performances can depend on the display sets, let us define an instance of the comparison by $\{\mathcal{S}_{train}, \mathcal{S}_{test}, \mathcal{C}\}$. Here $\mathcal{S}_{train}$ (resp. $\mathcal{S}_{test}$) is collection of display

---

[4]In the supplementary material, we also conducted a numerical experiment on an e-commerce dataset, and the result is consistent with experiment 1 on sushi data.

sets in the training (resp. test) data, and $\mathcal{C}$ is the number of clusters. In every instance, we split the data into 4000 participants in training and 1000 in testing. We generate empirical training (resp. testing) choice data by first enumerating all display sets in $\mathcal{S}_{train}$ (resp. $\mathcal{S}_{test}$) and then record every participant's favorite type of sushi for every display set in the training (resp. testing) data. Finally, we use the log-likelihood as performance metric. We will use term *explanation power* (resp. *prediction power*) to refer the performance metric evaluated on the training (resp. testing) data.

We use two configurations for $\mathcal{S}_{train}$ and $\mathcal{S}_{test}$, respectively. For $\mathcal{S}_{train}$, we take it to be either the full set or a collection of three randomly generated sets, which are realized to be $\{\{1, 3, 4, 7, 8, 10\}, \{2, 4, 5, 6, 8, 9\}, [10]\}$. For $\mathcal{S}_{test}$, we take it to be either the collection of all pairwise sets or all display sets with sizes at least two. Note that $\mathcal{S}_{train}$ has low variability and many elements in $\mathcal{S}_{test}$ do not appear in $\mathcal{S}_{train}$, making the experiment emphasizing more on the *generalization power*. We take $\mathcal{C} \in \{1, 2, 4, 6, 15\}$.

We summarize our results in Figure 1. We find that our method has favorable performance compared to MS and MNL across the settings. Intuitively, MS and MNL underperform in different ways. MS is vulnerable to the identifiability issue in the "choice-to-ranking" step, for which we provide theoretical understanding in Theorem 3. Our numerical experiments confirm the theoretical insight: We randomize over the extreme points of the polytope of solutions to (6), each of which leads to a valid output of the MS heuristic. Every box plot in Figure 1 contains a summary of the performances of those outputs. It is clear that the performances (both in training and testing) span wide ranges. In the meantime, MNL is vulnerable to both the model risk from its specific parametric form (reflected in the limited increase of explanation power when considering more clusters) and the sample risk of overfitting (reflected in the difference in performance in training vs. testing, especially when trained on the full display set only); see more details in the supplementary materials.

Figure 1: Comparison of Explanation and Prediction Power for Top-1 choice

In each panel, the x-axis represents the number of clusters, and the y-axis represents the log-likelihood metric.

**Experiment 2: Top-$k$ choice.** In this experiment, we perform a robustness check based on the criterion that a good model should return similar central rankings under different feedback structures, i.e., different $k$. Specifically, we conduct estimation on top-1, top-2 and top-3 ranked choices constructed from the first 10-sushi data set. The collection of display sets is taken to be all display sets with sizes at least $k$. We find that all three experiment instances produce the same estimated RMJ-implied central ranking $\hat{\pi} = (8, 5, 6, 3, 2, 1, 4, 9, 7, 10)$. Such robustness stands in contrast with other more naive methods, such as various versions of Borda count; see more details in the supplementary materials. We believe it presents evidence that our methodology is learning sensible information from the data.

**Experiment 3: 100 sushi types.** In this experiment, we wish to show how effective our method is for a relatively large number of items. In the data, each of the 5000 individuals indicates their top-10 choices out of 100 types of sushi. We use the LP-Rand-Pivot speed-up heuristic by [13] based on LP relaxation to train a single-cluster RMJ-based model. We bootstrap 10 times (each time drawing

10000 samples) and record the running times and optimality gaps in the Table 1. We find that we can obtain $< 2\%$ optimality gap within 5 minutes (excl. model building time).

Table 1: Computational Time and Optimality Gap on the 100-sushi data

|         | Model Building Time (min) | Model Solving Time (min) | Optimality Gap |
|---------|---------------------------|--------------------------|----------------|
| Average | 21.10                     | 4.20                     | 1.47%          |
| Max     | 21.19                     | 4.50                     | 1.79%          |

## 5  Conclusions

We identify a novel distance-based (Mallows-type) ranking model. It aggregates into simple probability distributions for top-$k$ subrankings among an arbitrary display set $S$. In addition, it facilitates effective parameter learning through the MLE formulation. This is the first distance-based ranking model with such properties (even for $k = 1$) to the best of our knowledge.

This ranking model can be used to model population preferences and provide a rationalizable way to model their ranked choices from given display sets. We demonstrate its practical value using real preference data. For example, under a mixture setting with only a few clusters, it shows promising prediction power, especially when there is a limited variety in the display sets.

For future steps, we believe our work can serve as the "infrastructure" for a range of business-related decision problems, such as new product introduction, crowdsourcing, and marketing research, among others.

## Acknowledgments and Disclosure of Funding

This work is supported by the Singapore Ministry of Education (MOE) Academic Research Fund (AcRF) Tier 1 [WBS Number: R-314-000-121-115]. The authors would like to thank René Caldentey and Christopher Thomas Ryan for earlier discussions of this problem, as well as Long Zhao and Xiaobo Li for their feedback. The authors are also very grateful to the Area Chair and reviewers for their careful reading of the paper and for the many helpful and constructive comments.

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
