# *On A Mallows-type Model For (Ranked) Choices*

## A Characterizing the Choice Probability (Theorem 1)

### A.1 Proofs of Intermediate Results (Lemmas 1 to 3)

**Proof of Lemma 1.** Let us first assign a weight to each ranking $\pi \in \Sigma$ and subranking $\pi_k \in \Sigma_k$ ($k \in \{0, \ldots, n\}$) by

$$w(\pi) := \frac{q^{d(\pi)}}{\psi(n, q)} \quad \text{and} \quad w(\pi_k) := \sum_{\pi \in \Sigma : \pi_k \subseteq \pi} w(\pi). \tag{7}$$

We note that on the one hand, $w \propto \lambda$. On the other hand, it is unclear *in priori* whether $w(\cdot)$ is the probability mass function since one needs to verify that $w(\cdot)$ is equipped with the right normalizing constant. We will first prove that $w(\pi_k)$ can also be written as $w(\pi_k) = q^{d(\pi_k)+L(\pi_k)} \cdot \frac{\psi(n-k,q)}{\psi(n,q)}$. Then we will show that for every $k \in [n]$, $\sum_{\pi_k \in \Sigma_k} w(\pi_k) = 1$. In other words, $w(\cdot) = \lambda(\cdot)$.

Step 1. Recall that for every $\pi_k \in \Sigma_k$, $d(\pi_k) = \sum_{i=1}^{k-1} \mathbb{I}\{\pi_k(i) > \pi_k(i+1)\}(n-i)$ and $L(\pi_k) := |\{x \in R^c(\pi_k) : x < \pi_k(k)\}|$. We prove from (7) that

$$w(\pi_k) = q^{d(\pi_k)+L(\pi_k)} \cdot \frac{\psi(n-k, q)}{\psi(n, q)}$$

by backward induction on $k$.

*Base step.* Suppose $k = n$. Pick an arbitrary $\pi_k \in \Sigma_k = \Sigma$. Because $\pi_k$ is a full ranking,

$$w(\pi_k) = q^{d(\pi_k)} \cdot \frac{1}{\psi(n, q)} \overset{(a)}{=} q^{d(\pi_k)+L(\pi_k)} \cdot \frac{\psi(n-k, q)}{\psi(n, q)}.$$

In the derivations above, part (a) is due to the facts that $\psi(n - k, q) = \psi(0, q) = 1$ and $L(\pi_k) = 0$ by definition.

*Inductive step.* Pick an arbitrary $K \in \{0, 1, \ldots, n-1\}$. Suppose our statement holds for every $k = K + 1, \ldots, n$. We want to show that our statement holds for $k = K$. In other words, pick an arbitrary $\pi_k \in \Sigma_k$. We wish to show that $\Sigma_k, w(\pi_k) = q^{d(\pi_k)+L(\pi_k)} \cdot \frac{\psi(n-k,q)}{\psi(n,q)}$.

We first claim that for every $\pi_{k+1} \in \Sigma_{k+1}$ such that $\pi_k \subseteq \pi_{k+1}$, $\pi_k(k) > \pi_{k+1}(k+1)$ if and only if $L(\pi_{k+1}) \leq L(\pi_k) - 1$. To see why, note that if $\pi_k(k) > \pi_{k+1}(k+1)$, $L(\pi_k) = L(\pi_{k+1}) + 1$. Otherwise, $L(\pi_k) \leq L(\pi_{k+1})$, and hence $L(\pi_k) - 1 < L(\pi_{k+1})$. As a consequence, $d(\pi_{k+1})$ can be expressed in terms of $d(\pi_k)$:

$$
\begin{aligned}
d(\pi_{k+1}) &= \sum_{i=1}^{k} \mathbb{I}\{\pi_k(i) > \pi_k(i+1)\} \cdot (n-i) \\
&= \underbrace{\sum_{i=1}^{k-1} \mathbb{I}\{\pi_k(i) > \pi_k(i+1)\} \cdot (n-i)}_{d(\pi_k)} + \underbrace{\mathbb{I}\{\pi_k(k) > \pi_k(k+1)\}}_{\mathbb{I}\{L(\pi_{k+1}) \leq L(\pi_k)-1\}} \cdot (n-k) \\
&= d(\pi_k) + \mathbb{I}\{L(\pi_{k+1}) \leq L(\pi_k) - 1\} \cdot (n-k).
\end{aligned}
$$

Moreover, given $\pi_k$, there is a one-to-one correspondence between $\pi_{k+1}$ and $k + 1$ and we use $\pi_k \oplus L^{-1}(i)(k+1)$ to represent the (unique) $\pi_{k+1}$ such that $\pi_k \subseteq \pi_{k+1}$ and $L(\pi_{k+1}) = i, (i \in \{0, 1, \ldots, n - k - 1\})$.

$$w\left(\pi_k\right) = \sum_{\pi \in \Sigma : \pi_k \subseteq \pi} w(\pi) = \sum_{\pi_{k+1} \in \Sigma_{k+1} : \pi_k \subseteq \pi_{k+1}} w\left(\pi_{k+1}\right)$$

$$= \sum_{i=0}^{n-k-1} w\left(\pi_k \oplus L^{-1}(i)(k+1)\right)$$

$$= \sum_{i=0}^{n-k-1} q^{d\left(\pi_k \oplus L^{-1}(i)(k+1)\right) + L\left(\pi_k \oplus L^{-1}(i)(k+1)\right)} \cdot \frac{\psi(n-k-1, q)}{\psi(n, q)}$$

$$= \frac{\psi(n-k-1, q)}{\psi(n, q)} \cdot \left( \sum_{i=0}^{L(\pi_k)-1} q^{d\left(\pi_k \oplus L^{-1}(i)(k+1)\right) + i} + \sum_{i=L(\pi_k)}^{n-k-1} q^{d\left(\pi_k \oplus L^{-1}(i)(k+1)\right) + i} \right)$$

$$= \frac{\psi(n-k-1, q)}{\psi(n, q)} \cdot \left( \sum_{i=0}^{L(\pi_k)-1} q^{d(\pi_k) + n - k + i} + \sum_{i=L(\pi_k)}^{n-k-1} q^{d(\pi_k) + i} \right)$$

$$= \frac{\psi(n-k-1, q)}{\psi(n, q)} \cdot q^{d(\pi_k)} \cdot q^{L(\pi_k)} \underbrace{\left( \sum_{i=n-k-L(\pi_k)}^{n-k-1} q^i + \sum_{i=0}^{n-k-1-L(\pi_k)} q^i \right)}_{\frac{\psi(n-k,q)}{\psi(n-k-1,q)}}$$

$$= q^{d(\pi_k) + L(\pi_k)} \cdot \frac{\psi(n-k, q)}{\psi(n, q)}.$$

Hence our statement holds for $k = K$, too, thus $w\left(\pi_k\right) = q^{d(\pi_k) + L(\pi_k)} \cdot \frac{\psi(n-k,q)}{\psi(n,q)}$ for any $k$.

Step 2. We verify that for every $k \in \{0, 1, \ldots, n\}$, $\sum_{\pi_k \in \Sigma_k} w(\pi_k) = 1$.

For all $k, k' \in \{0, 1, \ldots, n\}$, we invoke (7) and have $\sum_{\pi_k \in \Sigma_k} w(\pi_k) = \sum_{\pi \in \Sigma} w(\pi) = \sum_{\pi_{k'} \in \Sigma_{k'}} w(\pi_{k'})$. Therefore, take an arbitrary $k \in \{0, 1, \ldots, n\}$ and $k' = 2$,

$$\sum_{\pi_k \in \Sigma_k} w(\pi_k) = \sum_{\pi_2 \in \Sigma_2} w(\pi_2) = \sum_{i \in \{0, \ldots, n-1\}} w(\pi_1 \oplus i)$$

$$= \frac{\psi(n-1, q)}{\psi(n, q)} \cdot (1 + q + \ldots + q^{n-1})$$

$$= 1$$

Hence, the weight $w$ is the probability mass function $\lambda$. □

**Proof of Lemma 2.** By Lemma 1, we have

$$\frac{\lambda(\pi_k \oplus y)}{\lambda(\pi_k)} = \frac{q^{d(\pi_{k+1}) + L(\pi_{k+1})} \cdot \frac{\psi(n-k-1, q)}{\psi(n, q)}}{q^{d(\pi_k) + L(\pi_k)} \cdot \frac{\psi(n-k, q)}{\psi(n, q)}} = \frac{q^{d(\pi_k) + \mathbb{I}\{z > y\} \cdot (n-k) + L(\pi_{k+1})}}{q^{d(\pi_k) + L(\pi_k)}} \frac{1 - q}{1 - q^{n-k}}$$

$$= q^{\mathbb{I}\{z > y\} \cdot (n-k) + L(\pi_{k+1}) - L(\pi_k)} \frac{1 - q}{1 - q^{n-k}}.$$

By the definition of $L(\cdot)$,

$$\mathbb{I}\{z > y\} \cdot (n-k) + L(\pi_{k+1}) - L(\pi_k) = \begin{cases} \sum_{x \in R^c(\pi_k)} \mathbb{I}\{z < x \le y\} - 1 & \text{if } y > z \\ n - k - \sum_{x \in R^c(\pi_k)} \mathbb{I}\{y < x < z\} - 1 & \text{if } y < z \end{cases}.$$

The proof is finished. □

**Proof of Lemma 3.** We prove this lemma by backward induction on $k$.

*Base step.* Since when $k > n - M + 1$, $R(\pi_{k-1}) \cap S = \emptyset$ is not satisfied for all $\pi_{k-1} \in \Sigma_{k-1}$. We suppose $k = n - M + 1$. Pick an arbitrary $\pi_k = \pi_{k-1} \oplus z$ such that $R(\pi_{k-1}) \cap S = \emptyset$. Item $z$ is an item in $S$, so the choice outcome is deterministic. If $z = x_i$, the conditional choice probability is 1, and if $z \in S \setminus \{x_i\}$, the conditional choice probability is 0.

*Inductive step.* Pick an arbitrary $K \in \{1, \ldots, n - M\}$. Suppose our statement holds for every $k = K + 1, \ldots, n - M + 1$. We want to show that our statement holds for $k = K$.

Pick an arbitrary $\pi_k = \pi_{k-1} \oplus z$ such that $R(\pi_k) \cap S = \emptyset$. Similar to the base step, if $z = x_i$, the conditional choice probability is 1, and if $z \in S \setminus \{x_i\}$, the conditional choice probability is 0. When $z \notin S$, rename the items in $R^c(\pi_k)$ as $y_1, y_2, \ldots, y_{n-k}$ so that $y_1 < y_2 < \ldots < y_{n-k}$. We have the following decomposition.

$$Q := Pr(\{x_i|S\}|\pi_{k-1} \oplus z) = \sum_{j=1}^{n-k} Pr(\{x_i|S\}|\pi_k \oplus y_j) \cdot Pr(\pi_k \oplus y_j|\pi_{k-1} \oplus z)$$

Since $R(\pi_k) \cap S = \emptyset$, we have $\{x_1, x_2, \ldots, x_M\} \subseteq \{y_1, y_2, \ldots, y_{n-k}\}$, we use $\overline{x_i}$ to denote the relative position of item $x_i$ in these $n - k$ items, hence item $x_i$ is renamed as $y_{\overline{x_i}}$ under "y" notation. For example, if $\overline{x_i} = 4$, then there are 3 items with smaller indices (i.e., preferred under $\pi^* = e$) than item $x_i$ in $\{y_i\}_{i=1}^{n-k}$. Hence, we can rewrite $\{y_i\}_{i=1}^{n-k}$ as

$$\{y_1, \ldots, y_{\overline{x_1}-1}, y_{\overline{x_1}}, y_{\overline{x_1}+1}, \ldots, y_{\overline{x_i}-1}, y_{\overline{x_i}}, y_{\overline{x_i}+1}, \ldots, y_{n-k}\}.$$

By induction hypothesis, we have

$$Pr(\{x_i|S\}|\pi_k \oplus \pi_{k+1}(k+1)) = \begin{cases} \frac{q^{i-1}}{1+q+\ldots+q^{M-1}} & \text{if } \pi_{k+1}(k+1) \in \{y_a\}_{a=1}^{\overline{x_1}-1}, \\ \frac{q^{i-j-1}}{1+q+\ldots+q^{M-1}} & \text{if } \pi_{k+1}(k+1) \in \{y_a\}_{a=\overline{x_j}+1}^{\overline{x_{j+1}}-1} \text{ for } j \in \{1, \ldots, i-1\}, \\ 1 & \text{if } \pi_{k+1}(k+1) = y_{\overline{x_i}}, \\ \frac{q^{M+i-j-1}}{1+q+\ldots+q^{M-1}} & \text{if } \pi_{k+1}(k+1) \in \{y_a\}_{a=\overline{x_j}+1}^{\overline{x_{j+1}}-1} \text{ for } j \in \{i, \ldots, M-1\}, \\ \frac{q^{i-1}}{1+q+\ldots+q^{M-1}} & \text{if } \pi_{k+1}(k+1) \in \{y_a\}_{a=\overline{x_M}+1}^{n-k}, \\ 0 & \text{if } \pi_{k+1}(k+1) = y_{\overline{x_j}} \text{ for } x_j \in S \setminus \{x_i\}. \end{cases}$$

By Lemma 2, we know $Pr(\pi_k \oplus y_j|\pi_{k-1} \oplus z)$. Finally, we can get

$$Q = \begin{cases} \frac{q^{M-p(z|S)+i-1}}{1+q+\ldots+q^{M-1}} & \text{if } z > x_i \text{ and } z \notin S, \\[2mm] \frac{q^{i-p(z|S)-1}}{1+q+\ldots+q^{M-1}} & \text{if } z < x_i \text{ and } z \notin S. \end{cases}$$

Hence our statement holds for $k = K$, too, thus finishing the proof. $\square$

## A.2 Putting Things Together for Theorem 1

**Proof of Theorem 1.** Name all items as $\{y_z\}_{z=1}^n$ such that $y_1 < y_2 < \ldots < y_n$. We have

$$Pr(\{x_i|S\}) = \sum_{z=1}^n Pr(\{x_i|S\}|(y_z)) \cdot \lambda((y_z))$$

Use $\overline{x_j}$ to denote the position of item $x_j$ in the universe, hence item $x_j$ is renamed as $y_{\overline{x_j}}$ under "y" notation. By Lemma 3, we have the following equation:

$$Pr(\{x_i|S\}|(y_z)) = \begin{cases} \frac{q^{i-1}}{1+q+\ldots+q^{M-1}} & \text{if } y_z \in \{y_a\}_{a=1}^{\overline{x_1}-1}, \\ \frac{q^{i-j-1}}{1+q+\ldots+q^{M-1}} & \text{if } y_z \in \{y_a\}_{a=\overline{x_j}+1}^{\overline{x_{j+1}}-1} \text{ for } j \in \{1, \ldots, i-1\}, \\ 1 & \text{if } y_z = y_{\overline{x_i}}, \\ \frac{q^{M+i-j-1}}{1+q+\ldots+q^{M-1}} & \text{if } y_z \in \{y_a\}_{a=\overline{x_j}+1}^{\overline{x_{j+1}}-1} \text{ for } j \in \{i, \ldots, M-1\}, \\ \frac{q^{i-1}}{1+q+\ldots+q^{M-1}} & \text{if } y_z \in \{y_a\}_{a=\overline{x_M}+1}^{n}, \\ 0 & \text{if } y_z = y_{\overline{x_j}} \text{ for } x_j \in S \setminus \{x_i\}. \end{cases}$$

By Lemma 1, we have $\lambda((y_z)) = \frac{q^{z-1}}{1+\ldots+q^{n-1}}$. After calculation, we get $Pr(\{x_i|S\}) = \frac{q^{i-1}}{1+q+\ldots+q^{M-1}}$.
$\square$

# B Consistency of MLE for OAM (Theorem 2)

## B.1 Main Body of the Proof

**Proof of Theorem 2.** First fix the underlying parameters of the RMJ-based ranking model $(\pi^*, q^*)$. Let the choice data $H_T = (S_1, x_1, \ldots, S_T, x_T)$ be given, where $S_t$ is the display set shown to the $t$th participant and $x_t$ is his/her choice. Invoking Proposition 3 of [13], the MLE problem for OAM, the choice model induced by the RMJ-based ranking distribution, can be written as

$$(\hat{\pi}, \hat{q}) \in \operatorname{argmax}_{\pi, q} \quad \sum_{t=1}^{T} \log\left(\frac{1-q}{1-q^{|S_t|}}\right) + \log q \sum_{(i,j):i \neq j} \mathbb{I}\{j \succ_\pi i\} w_{ij},$$

where

$$w_{ij} := \sum_{t=1}^{T} \mathbb{I}\Big\{\{i, j\} \subseteq S_t \text{ and } x_t = i\Big\}$$

is the number of times that both items $i$ and $j$ are displayed and item $i$ is chosen (among the $T$ samples). Furthermore, recall that $\hat{\pi}$ can be obtained from the integer programming formulation (5), which is further equivalent to the following formulation by substituting the relation $x_{ij} + x_{ji} = 1$:

$$\{\hat{x}_{ij} : i < j\} \in \operatorname{argmin}_{\{x_{ij}:i<j\}} \quad \sum_{(i,j):i<j} (w_{ji} - w_{ij}) x_{ij}$$

$$\text{s.t.} \quad \begin{array}{ll} x_{ij} + x_{jr} - x_{ir} \leq 1 & \forall \, 1 \leq i < j < r \leq n \\ x_{ij} \in \{0, 1\} & \forall \, 1 \leq i < j \leq n \end{array} \tag{8}$$

The ranking $\hat{\pi}$ is obtained by letting $i \succ_\pi j$ if and only if $\hat{x}_{ij} = 1$. Given the solution of $\hat{\pi}$, the estimator $\hat{q}$ is obtained by the one-dimensional convex optimization

$$\hat{\alpha} \in \operatorname{argmin}_{\alpha \in (0, +\infty)} \quad L_T(\alpha) := \alpha \sum_{t=1}^{T} \sum_{x \in S_t} \mathbb{I}\{x \succ_{\hat{\pi}} x_t\} + \sum_{t=1}^{T} \log \sum_{j=0}^{|S_t|-1} e^{-j\alpha} \tag{9}$$

so that $\hat{\alpha} = -\log \hat{q}$.

We break the rest of the proof into two parts: the "if" part and the "only if" part.

The "if" part. Let $\mathcal{S}_\infty$ be the collection of display sets that are displayed infinite times. Suppose for every pair of items $\{i, j\}$ is covered infinitely many times. That is, there exists a display set $S \in \mathcal{S}_\infty$ such that $\{i, j\} \subseteq S$. We wish to show that $(\hat{\pi}, \hat{q}) \to (\pi^*, q^*)$ almost surely as the sample size $T \to \infty$.

We first claim that $\hat{\pi} \to \pi^*$ almost surely. Without loss of generality, assume $\pi^* = e$, the identity ranking. In other words, we wish to show that with probability one, there exists $\tau$ such that for all $T \geq \tau$, the unique solution to (8) is $\{x_{ij} = 1, i < j\}$. Pick an arbitrary pair of items $i, j$ such that $i < j$. Let $N_{ij} := \sum_{t=1}^{T} \mathbb{I}\{\{i, j\} \subseteq S_t\}$ be the number of times that both items $i$ and $j$ are displayed. Because both $i$ and $j$ are covered by some $S \in \mathcal{S}_\infty$, we have $N_{ij} \to \infty$ as $T \to \infty$. Note that invoking the choice probabilities in (4), OAM is a $q^*$-separable choice model, i.e., for every display set $S$ such that $\{i, j\} \subseteq S$, $\Pr(\{j|S\}) \leq q \Pr(\{i|S\})$; see [13] for more details. Since the choices are generated independently conditional the display sets, we invoke the law of large numbers and conclude that $w_{ij} \to \infty$, $w_{ji} \to \infty$, and $w_{ji}/w_{ij} \to q'$ for some $q' \leq q$ as the sample size $T \to \infty$. Furthermore, since there are only a finite number of pairs, with probability one, there exists $\tau$ such that for all $T \geq \tau$,

$$w_{ji} - w_{ij} < 0 \quad \text{for all } i < j.$$

In that case, it is straightforward to see that the unique solution to (8) is $\{x_{ij} = 1, i < j\}$. The "if" part is hence completed by noting the result below. Its proof resembles the standard argument for consistency of MLE except for a few technical differences, such as allowing for an arbitrary display set offering process (which leads to not necessarily i.i.d choice data) and non-compactness of the range of $\alpha$. We provide the proof details in Appendix B.2.

**Lemma 4** $\hat{q} \to q^*$ *almost surely.*

The "only if" part. Suppose there exists a pair of items $\{i, j\}$ that is not covered infinitely many times. We wish to show that for some underlying parameter $(\pi^*, q^*)$, $(\hat{\pi}, \hat{q}) \not\to (\pi^*, q^*)$ with positive probability as the sample size $T \to \infty$.

Through a relabeling argument, assume $i = n - 1$ and $j = n$ without loss of generality. That is, the items $\{n - 1, n\}$ are only displayed together finitely many times. Suppose $\pi^* = e$ is the ground truth ranking. We claim that it does not hold that $\hat{\pi} \to e$ almost surely as $T \to \infty$.

The rest of proof consists of two steps. First, note that the items $\{n - 1, n\}$ are only displayed finitely many times. Therefore, with a positive probability, there exists $\bar{T} > 0$ such that $w_{n-1,n} \leq w_{n,n-1}$ for all $T \geq \bar{T}$. (In particular, if $\{n - 1, n\}$ is not covered *at all*, we have $w_{n-1,n} = w_{n,n-1} = 0$ for all $T \geq 1$.)

Second, for the sake of contradiction, suppose $\hat{\pi} \to e$ almost surely as $T \to \infty$. Then with probability one, there exists $\tau < +\infty$ so that for all $T \geq \tau$, $\hat{\pi} = e$. It further implies that $\hat{x} := \{\hat{x}_{ij} = 1, \text{ for all } i < j\}$ is an optimal solution to (8). However, notice that if $w_{n-1,n} \leq w_{n,n-1}$, the solution $\tilde{x}$ defined by

$$\tilde{x}_{ij} = \begin{cases} 1, & \text{if } i < j \text{ and } (i, j) \neq (n - 1, n) \\ 0, & \text{if } i = n - 1 \text{ and } j = n \end{cases}$$

weakly decreases the objective function value. It is also feasible because $\tilde{x}$ corresponds to the ranking $\tilde{\pi} := (1, 2, \ldots, n - 2, n, n - 1)$ (i.e., the ranking obtained by swapping the rankings of the bottom two ranked items of $e$). Therefore, $\tilde{x}$ is also an optimal solution to (8). Invoking the first step, we know that with positive probability, both $e$ and $\tilde{\pi}$ are MLEs for all $T \geq \max\{\tau, \bar{T}\}$. Note that the tie-breaking rule for MLE cannot be item-specific (for otherwise $\tilde{\pi}$ cannot be correctly identified if it is the ground truth ranking). Therefore, with positive probability, no item-blind tie-breaking rule for MLE can differentiate between items $n - 1$ and $n$ and thus between rankings $e$ and $\tilde{\pi}$. As a result, we conclude that it cannot hold that $\hat{\pi} \to e$ almost surely as $T \to \infty$. $\square$

## B.2 Proof of Auxiliary Results (Lemma 4)

**Proof of Lemma 4.** Let $\alpha^* := -\log q^* \in (0, +\infty)$ and recall that $\hat{\alpha}$ is solution to (9). It suffices to show that $\hat{\alpha} \to \alpha^*$ almost surely. We employ a pathwise analysis throughout the proof.

Given display set $S$ and item $x \in S$, let $N_S := \sum_{t=1}^{T} \mathbb{I}\{S_t = S\}$ and $N_S^x := \sum_{t=1}^{T} \mathbb{I}\{S_t = S \text{ and } x_t = x\}$ be the number of times that $S$ is displayed and $x$ is chosen out of $S$, respectively. For $i \in \{1, \ldots, |S|\}$, let $\pi_S(i)$ be the $i^{th}$ most preferred item within display set under ranking $\pi^*$. In other words, if $\pi_S(i) = y$, then $\sum_{x \in S} \mathbb{I}\{x \succ_{\pi^*} y\} = i - 1$. Finally, let us introduce

$$L_T^S(\alpha) := \frac{1}{N_S}\left(\alpha \sum_{t:S_t=S} \sum_{x \in S} \mathbb{I}\{x \succ_{\pi^*} x_t\} + \sum_{t:S_t=S} \log \sum_{j=0}^{|S|-1} e^{-j\alpha}\right)$$

$$= \alpha \sum_{i=1}^{|S|} \frac{N_S^{\pi_S(i)}}{N_S}(i - 1) + \log \sum_{j=0}^{|S|-1} e^{-j\alpha}$$

to be the (scaled) partial log likelihood loss function when $\hat{\pi} = \pi^*$ and only display set $S$ is considered. Since the choices are generated independently conditional the display sets, we invoke the choice probabilities in (4) as well as the law of large numbers and conclude that with probability one, for all $S \in \mathcal{S}_\infty$ and $\alpha \in (0, +\infty)$,

$$L_T^S(\alpha) \to L_\infty^S(\alpha) := \alpha \sum_{i=1}^{|S|} \frac{e^{-(i-1)\alpha^*}}{\sum_{j=0}^{|S|-1} e^{-j\alpha^*}}(i - 1) + \log \sum_{j=0}^{|S|-1} e^{-j\alpha}$$

$$= \alpha \frac{\sum_{i=1}^{|S|-1} i e^{-i\alpha^*}}{\sum_{j=0}^{|S|-1} e^{-j\alpha^*}} + \log \sum_{j=0}^{|S|-1} e^{-j\alpha}.$$

In fact, the convergence is also locally uniform: for all $M > 0$,

$$\sup_{\alpha \in (0,M]} |L_T^S(\alpha) - L_\infty^S(\alpha)| = \sup_{\alpha \in (0,M]} \alpha \sum_{i=1}^{|S|} \left| \frac{N_S^{\pi_S(i)}}{N_S} - \frac{e^{-(i-1)\alpha^*}}{\sum_{j=0}^{|S|-1} e^{-j\alpha^*}} \right| (i-1)$$

$$\leq M \sum_{i=1}^{|S|} \left| \frac{N_S^{\pi_S(i)}}{N_S} - \frac{e^{-(i-1)\alpha^*}}{\sum_{j=0}^{|S|-1} e^{-j\alpha^*}} \right| (i-1) \to 0.$$

as sample size $T \to \infty$. Therefore, with probability one, $L_T^S(\cdot) \to L_\infty^S(\cdot)$ uniformly on the set $(0, M]$.

In addition, it is straightforward to verify that for every $S$, $L_\infty^S(\cdot)$ is strictly convex and attains its unique minimum at $\alpha^*$. In fact, one can verify that the first order condition is

$$(L_\infty^S)'(\alpha) = 0 \Rightarrow \frac{\sum_{i=1}^{|S|-1} i e^{-i\alpha^*}}{\sum_{j=0}^{|S|-1} e^{-j\alpha^*}} = \frac{\sum_{i=1}^{|S|-1} i e^{-i\alpha}}{\sum_{j=0}^{|S|-1} e^{-j\alpha}},$$

which implies that $\operatorname{argmin}_\alpha L_\infty^S(\alpha) = \alpha^*$. Since $L_T^S(\cdot) \to L_\infty^S(\cdot)$ uniformly on the set $(0, M]$ for every $M > 0$, we further conclude that with probability one and for an arbitrarily small $\epsilon > 0$, there exists $\tau_2$ such that for all $T \geq \tau_2$,

$$L_T(\alpha) = \sum_{S \in \mathcal{S}_\infty} N_S L_T^S(\alpha) + O(1)$$

attains its minimum in $[\alpha^* - \epsilon, \alpha^* + \epsilon]$. In other words, $\hat{\alpha} \in [\alpha^* - \epsilon, \alpha^* + \epsilon]$. $\square$

## C  Inconsistency of the Mallows Smoothing Heuristic (Theorem 3)

### C.1  Overview

The Mallows Smoothing heuristic ([4]) consists of a two-step process.

Step 1: ("Choice to Ranking") This step takes the empirical choice probabilities $\{\hat{\mathrm{Pr}}(x|S)\}$ as input and produces a distribution over rankings $\hat{\lambda}$ as output. The goal of this step is to find $\hat{\lambda}$, under which the aggregated choice probabilities match the empirical choice probabilities. Formally, let $\hat{\mathcal{S}}$ be the collection of display sets that have appeared in the data. This step corresponds to finding a solution $\hat{\lambda}$ to the following feasibility problem:[5]

$$\min_{\hat{\lambda}} \quad 0$$
$$\sum_{\tilde{\pi} \in \Sigma} \hat{\lambda}(\tilde{\pi}) \, \mathbf{I}(x, \tilde{\pi}, S) = \hat{\mathrm{Pr}}(x|S), \quad \text{for all } S \in \hat{\mathcal{S}} \text{ and } x \in S$$
$$\sum_{\tilde{\pi} \in \Sigma} \hat{\lambda}(\tilde{\pi}) = 1 \tag{10}$$
$$\hat{\lambda}(\pi) \geq 0, \qquad\qquad \text{for all } \pi \in \Sigma.$$

Step 2: ("Smoothing") This step takes a distribution over rankings $\hat{\lambda}$ as input and produces the Mallows distribution parameters $(\hat{\pi}, \hat{q})$ as output. The goal of this step is to find a Mallows distribution that fits the $\hat{\lambda}$ distribution produced by the previous step.

Formally, recall that the Kendall's Tau distance between two rankings $\pi, \tilde{\pi}$ is given by $d_K(\pi, \tilde{\pi}) = \sum_{x<y} \mathbb{I}\{(\pi^{-1}(x) - \pi^{-1}(y)) \cdot (\tilde{\pi}^{-1}(x) - \tilde{\pi}^{-1}(y)) < 0\}$. The estimated central ranking $\hat{\pi}$ of the Mallows distribution is obtained from the following ranking aggregation problem:

$$\hat{\pi} \in \operatorname{argmin}_\pi \sum_{\tilde{\pi}} \hat{\lambda}(\tilde{\pi}) \cdot d_K(\pi, \tilde{\pi}). \tag{11}$$

---

[5]If the system above is not feasible, then solve a "soft" version of it by minimizing a loss function of residues; see [7] for more details.

The dispersion parameter is obtained from solving the following convex problem:

$$\hat{q} = \text{argmin}_q \, q \cdot \sum_{\tilde{\pi}} \hat{\lambda}(\tilde{\pi}) \cdot d_K(\hat{\pi}, \tilde{\pi}) + T \log \psi(n, q).$$

Note that in system (10), the number of variables is on the order of $\Omega(n!)$ while the number of constraints is on the order of $O(n2^n)$, which is much smaller. Therefore, whenever feasible, the solution forms a high dimensional polytope.

Roughly speaking, we will show that even when all display sets with sizes at least three are displayed infinite many times, the Mallows model parameters cannot be identified from the MS heuristic. That is, we suppose that infinite choice data is generated from a Mallows model with the central ranking to be $e$. However, we can construct a ranking distribution $\tilde{\lambda}$ that solves (10) so that when we use it as input to problem (11), we find a solution $\tilde{\pi}$ that is different from $e$. That is, a *wrong* model parameter can be the output of the MS heuristic.

## C.2 Main Body of the Proof

**Proof of Theorem 3.** Pick $q \in (0, 1)$. Let $\lambda^e$ be the p.m.f. of the Mallows ranking model with the central ranking to be the identity ranking $e$ and the dispersion parameter to be $q$. Let $\{\Pr^e(x|S) : |S| \geq 3, x \in S\}$ be the collection of associated choice probabilities for display sets with sizes at least three, formally defined as $\Pr^e(x|S) := \sum_{\tilde{\pi} \in \Sigma} \lambda^e(\tilde{\pi}) \, \mathbf{I}(x, \tilde{\pi}, S)$ for all $S$ such that $|S| \geq 3$ and $x \in S$.

We claim that we can construct $\tilde{\lambda} \in \Delta(\Sigma)$ that satisfies two properties simultaneously.

1. First, it solves (10) when taking $\{\Pr^e(x|S) : |S| \geq 3, x \in S\}$ as input. That is, $\tilde{\lambda} \in \Delta(\Sigma)$ satisfies

$$\sum_{\tilde{\pi} \in \Sigma} \tilde{\lambda}(\tilde{\pi}) \, \mathbf{I}(x, \tilde{\pi}, S) = \Pr^e(x|S), \text{ for all } S \text{ such that } |S| \geq 3 \text{ and } x \in S. \quad (12)$$

2. Second, if we take $\tilde{\lambda}$ as input to (11), we obtain $\tilde{\pi} := (1, 2, \ldots, n-2, n, n-1) \neq e$ as solution. That is,

$$(1, 2, \ldots, n-2, n, n-1) \in \text{argmin}_\pi \sum_{\tilde{\pi}} \tilde{\lambda}(\tilde{\pi}) \cdot d_K(\pi, \tilde{\pi}). \quad (13)$$

Our construction is based on two observations. First, we invoke Theorem 3.7 by [7] and know that under the current display set setup, (10) can only identify a ranking up to its first $n-2$ positions. As a consequence, as long as $\tilde{\lambda}$ satisfies

$$\sum_{\tilde{\pi} \in \Sigma} \tilde{\lambda}(\tilde{\pi}) \, \mathbb{I}\{\pi_{n-2} \subseteq \tilde{\pi}\} = \sum_{\tilde{\pi} \in \Sigma} \lambda^e(\tilde{\pi}) \, \mathbb{I}\{\pi_{n-2} \subseteq \tilde{\pi}\}, \quad \text{for all } \pi_{n-2} \in \Sigma_{n-2}, \quad (14)$$

it also satisfies (12).

Second, we observe that problem (11) only depends on the *pairwise* choice probabilities associated with its input ranking distribution. This observation is formalized as the result below, and we present its proof in Appendix C.3.

**Lemma 5** *Pick* $\hat{\lambda} \in \Delta(\Sigma)$ *and let* $P_{x, \{x,y\}} := \sum_{\tilde{\pi}} \hat{\lambda}(\tilde{\pi}) \mathbb{I}\{\tilde{\pi}^{-1}(x) < \tilde{\pi}^{-1}(y)\}$ *be its associated pairwise probability of choosing item* $x$ *out of* $\{x, y\}$. *Then*

$$\text{argmin}_\pi \sum_{\tilde{\pi}} \hat{\lambda}(\tilde{\pi}) \cdot d_K(\pi, \tilde{\pi}) = \text{argmin}_\pi \sum_{x < y} \mathbb{I}\{\pi^{-1}(x) < \pi^{-1}(y)\} \cdot (1 - 2P_{x, \{x,y\}}).$$

Let $\tilde{P}_{x, \{x,y\}} := \sum_{\tilde{\pi}} \tilde{\lambda}(\tilde{\pi}) \mathbb{I}\{\tilde{\pi}^{-1}(x) < \tilde{\pi}^{-1}(y)\}$ As a quick consequence of the result above, a sufficient condition of (13) is

$$\tilde{P}_{x, \{x,y\}} > 1/2 \text{ for all } (x, y) \in \{(x, y) : x < y, (x, y) \neq (n-1, n)\} \text{ and } \tilde{P}_{n-1, \{n-1, n\}} < 1/2. \quad (15)$$

In order to construct $\tilde{\lambda}$ that satisfies both (14) and (15) (and therefore, fulfills out claim), let us classify $\Sigma$ into 3 groups based on its top-$(n-2)$ elements.

**Group 1:** Under $\pi_{n-2}$, item $(n-1)$ is preferred to item $n$. That is, either (i) $\{n-1,n\} \subseteq R(\pi_{n-2})$ and $\pi_{n-2}^{-1}(n-1) < \pi_{n-2}^{-1}(n)$, or (ii) $(n-1) \in R(\pi_{n-2})$ but $n \in R^c(\pi_{n-2})$.

**Group 2:** Under $\pi_{n-2}$, item $n$ is preferred to item $(n-1)$. That is, either (i) $\{n-1,n\} \subseteq R(\pi_{n-2})$ and $\pi_{n-2}^{-1}(n-1) > \pi_{n-2}^{-1}(n)$, or (ii) $n \in R(\pi_{n-2})$ but $(n-1) \in R^c(\pi_{n-21})$.

**Group 3:** Under $\pi_{n-2}$, items $n$ and $(n-1)$ are incomparable. That is $\{n-1,n\} \subseteq R^c(\pi_{n-2})$.

We explicitly construct $\tilde{\lambda}$ as the following:

$$\tilde{\lambda}(\pi) = \begin{cases} \lambda^e(\pi) & \text{if } \pi \text{ belongs to Group 1 or 2} \\ 0 & \text{if } \pi \in \text{ Group 3 and } (n-1) \succ_\pi n \\ \lambda^e(\pi_{n-2} \oplus (n-1,n)) + \lambda^e(\pi_{n-2} \oplus (n,n-1)) & \text{if } \pi \in \text{ Group 3 and } n \succ_\pi (n-1) \end{cases}$$

In other words, $\tilde{\lambda}$ is obtained from $\lambda^e$ by "transporting" weights in favor of item $n$ over item $(n-1)$ when both of those items are ranked at the bottom two. Since the top-$(n-2)$ rankings are not disturbed, (14) is satisfied by construction.

The rest of proof is devoted to verifying (15). Let us use $\lambda = \lambda^e$ and its associated pairwise choice probability $P_{x,\{x,y\}}$ for shorthand notation. Since only items $(n-1)$ and $n$ are swapped when they are at ranked at the bottom, we have $\tilde{P}_{x,\{x,y\}} = P_{x,\{x,y\}}$ for all $(x,y) \in \{(x,y) : x < y, (x,y) \neq (n-1,n)\}$. Therefore, $\tilde{P}_{x,\{x,y\}} > 1/2$ for all $(x,y) \in \{(x,y) : x < y, (x,y) \neq (n-1,n)\}$. Denote the sum of probabilities (under the Mallows model) of top-$(n-2)$ rankings in Group 1 as $\lambda(\text{Group 1}) := \sum_{\tilde{\pi}} \lambda(\tilde{\pi}) \mathbb{I}\{\tilde{\pi} \in \text{ Group 1}\}$. Given the construction rule, we have $\tilde{P}_{n-1,\{n-1,n\}} = \lambda(\text{Group 1})$. Therefore, to verify (15), it suffices to show

$$\lambda(\text{Group 1}) < \tfrac{1}{2}.$$

We can classify all top-$(n-2)$ rankings into four classes. Given a top-$(n-2)$ ranking, we call an item a *head item* if it ranks in top-$(n-2)$ and a *tail item* otherwise. We use $\lambda_{n-2}(\cdot)$ to denote the pmf of a top-$(n-2)$ ranking under Mallows in the following proof.

1. **Class A**: Item $(n-1)$ and item $n$ are both tail items. Denote the sum of probabilities of top-$(n-2)$ rankings in Class A as A, i.e., $A := \Pr(\text{Class A})$. Denote the last two items in a top$-(n-2)$ ranking as $x_1$ and $x_2$, and we can further partition Class A into 2 subclasses.

   (a) **Class A1**: $x_1 < x_2$. $A1 := \Pr(\text{Class A1})$.
   (b) **Class A2**: $x_1 > x_2$. $A2 := \Pr(\text{Class A2})$.

   There is a bijection between Class A1 and Class A2. By the closed form of the pmf of a top-$(n-2)$ ranking in Lemma 1 of [10], for each bijection pair $(a_1, a_2)$, $\lambda_{n-2}(a_2) = q \cdot \lambda_{n-2}(a_1)$. As a result, A2 = q · A1. Further, we have $A1 = \frac{A}{1+q}, A2 = \frac{qA}{1+q}$.

2. **Class B**: Item $(n-1)$ is a head item and item $n$ is a tail item. $B := \Pr(\text{Class B})$.
   We can further partition Class B into n-2 subclasses by the position of item $n-1$. For example, if the position is $m$, then the subclass is Class Bm and $Bm := \Pr(\text{Class Bm})$.
   There is a bijection between Class A and each subclass. For subclass Bm, for each bijection pair $(b_m, a)$, $\lambda_{n-2}(b_m) = q^m \cdot \lambda_{n-2}(a)$, then $Bm = q^m \cdot A$. Hence, we have $B = \sum_{m=1}^{n-2} q^m \cdot A$.

3. **Class C**: Item $n$ is a head item and item $(n-1)$ is a tail item. $C := \Pr(\text{Class C})$.
   There is a bijection between Class B and Class C. We have $C = q \cdot B$

4. **Class D**: Item $(n-1)$ and item $n$ are both head items. $D := \Pr(\text{Class D})$.
   This case is the most sophisticated. First, for a top-$(n-2)$ ranking $\pi_{n-2}$, assume $\pi_{n-2}^{-1}(n-1) = i$ and $\pi_{n-2}^{-1}(n) = j$. We can partition Class D into two subclasses.

   (a) **Class D1**: $i < j$. $D1 := \Pr(\text{Class D1})$. and $D1_{(i,j)} := \Pr(\text{Class D1} : \pi_{n-2}^{-1}(n-1) = i, \pi_{n-2}^{-1}(n) = j\})$.
   (b) **Class D2**: $i > j$. $D2 := \Pr(\text{Class D2})$. and $D2_{(i,j)} := \Pr(\text{Class D2} : \pi_{n-2}^{-1}(n-1) = i, \pi_{n-2}^{-1}(n) = j\})$.

There is a bijection between Class D1 and Class A1. $D1_{(i,j)} = q^{2n-1-i-j} \cdot A1$.

There is also a bijection between Class D1 and Class D2. $D2_{(i,j)} = q \cdot D1_{(i,j)}$.

Finally, the probability of every class can be expressed by $A$ and their sum is 1, so we get $A = \frac{(1-q)(1-q^2)}{(1-q^{n-1})(1-q^n)}$

Class B and Class D1 make up group 1, Class C and Class D2 make up group 2, and group 3 only contains Class A. Hence, $\Pr(\text{Group } 1) = \frac{1-A}{1+q}$ and it is not larger than 1/2 when $n$ and $q$ satisfy

$$\left(1 - \frac{(1-q)(1-q^2)}{(1-q^{n-1})(1-q^n)}\right) \frac{1}{1+q} < \frac{1}{2},$$

which simplifies to

$$\mathcal{F}_n(q) := 1 + q^{n-1} + q^n - 2q^2 - q^{2n-1} > 0.$$

Note that $\mathcal{F}_n(0) = 1 > 0, \mathcal{F}_n(1) = 0$, and $\mathcal{F}'_n(1) = (n-1) + n - 4 - (2n-1) < 0$. Therefore, we conclude that for all $n$, $\mathcal{F}_n(q) > 0$ either when $q$ is sufficiently close to 0 or 1. That finishes the proof.

□

### C.3   Proof of Auxiliary Results (Lemma 5)

**Proof of Lemma 5.**

$$\operatorname{argmin}_\pi \sum_{\tilde\pi} \hat\lambda(\tilde\pi) \cdot d_K(\pi, \tilde\pi)$$

$$= \operatorname{argmin}_\pi \sum_{\tilde\pi} \hat\lambda(\tilde\pi) \sum_{x<y} \mathbb{I}\{\left(\pi^{-1}(x) - \pi^{-1}(y)\right) \cdot \left(\tilde\pi^{-1}(x) - \tilde\pi^{-1}(y)\right) < 0\}$$

$$= \operatorname{argmin}_\pi \sum_{\tilde\pi} \hat\lambda(\tilde\pi) \sum_{x<y} [\mathbb{I}\{\tilde\pi^{-1}(x) < \tilde\pi^{-1}(y)\} + \mathbb{I}\{\pi^{-1}(x) < \pi^{-1}(y)\} \cdot (1 - 2\mathbb{I}\{\tilde\pi^{-1}(x) < \tilde\pi^{-1}(y)\})]$$

$$= \operatorname{argmin}_\pi \sum_{x<y} \mathbb{I}\{\pi^{-1}(x) < \pi^{-1}(y)\} \cdot \sum_{\tilde\pi} \hat\lambda(\tilde\pi)(1 - 2\mathbb{I}\{\tilde\pi^{-1}(x) < \tilde\pi^{-1}(y)\})$$

$$= \operatorname{argmin}_\pi \sum_{x<y} \mathbb{I}\{\pi^{-1}(x) < \pi^{-1}(y)\} \cdot (1 - 2\sum_{\tilde\pi} \hat\lambda(\tilde\pi)\mathbb{I}\{\tilde\pi^{-1}(x) < \tilde\pi^{-1}(y)\})$$

$$= \operatorname{argmin}_\pi \sum_{x<y} \mathbb{I}\{\pi^{-1}(x) < \pi^{-1}(y)\} \cdot (1 - 2P_{x,\{x,y\}})$$

□

## D   On the Ranked Choice Probabilities and Model Estimation for $k \geq 1$ (Theorems 4 and 5)

**Proof of Theorem 4.** Without loss of generality, suppose the display set is given by $S = \{x_1, \ldots, x_M\}$ such that $x_1 < x_2 < \cdots < x_M$ for some $M \geq 2$. We prove this theorem by forward induction on $k$ (i.e., the length of the ranked list).

*Base step.* Suppose $k = 1$. Given $\pi_1 = (z)$ where $z \in S$, we have $d_S(\pi_1) = 0$, $L_S(\pi_1) = |\{x \in S \setminus \{z\} : x < z\}|$ the relative ranking of $z$ in display set $S$, and $\frac{\psi(|S|-k,q)}{\psi(|S|,q)} = 1 + q + \cdots + q^{|S|-1}$. $q^{d_S(\pi_1)+L_S(\pi_1)} \cdot \frac{\psi(|S|-1,q)}{\psi(|S|,q)}$ is exactly the choice probability in Theorem 1.

*Inductive step.* Pick an arbitrary $K \in \{2, \ldots, M\}$. Suppose our statement holds for every $k = 1, \ldots, K-1$. We want to show that our statement holds for $k = K$.

Pick an arbitrary $\pi_k = (x_1^k, x_2^k, \ldots, x_{k-1}^k, x_k^k) = \pi_{k-1} \oplus x_k^k$. We have

$$Pr(\pi_k|S) = Pr(\pi_{k-1}|S) \cdot Pr(\{x_k^k|S \setminus R(\pi_{k-1})\}|\pi_{k-1} \text{ ranks the top-}(k-1) \text{ among } S)$$

Denote a subset $\Sigma'_{m-1}$, such that $\forall \pi_{m-1} \in \Sigma'_{m-1}, \pi_{k-2}$ ranks top-$(k-2)$ among $S$ in $\pi_{m-1}$ and $x \notin \pi_{m-1} \forall x \in S \backslash R(\pi_{k-2})$. Denote $c := Pr(\{x_k^k | S \backslash R(\pi_{k-1})\} | \pi_{m-1} \oplus x_{k-1}^k)$.

The second part of the RHS can be further decomposed as follows,

$$Pr(\{x_k^k | S \backslash R(\pi_{k-1})\} | \pi_{k-1} \text{ ranks the top-}(k-1) \text{ among } S)$$

$$= \sum_{m=k-1}^{n-M+k-1} \sum_{\pi_{m-1} \in \Sigma'_{m-1}} Pr(\{x_k^k | S \backslash R(\pi_{k-1})\} | \pi_{m-1} \oplus x_{k-1}^k)$$

$$\cdot Pr(\pi_{m-1} \oplus x_{k-1}^k | \pi_{k-1} \text{ ranks the top-}(k-1) \text{ among S})$$

$$= c \cdot \sum_{k=2}^{n-M+k-1} \sum_{\pi_{m-1} \in \Sigma'_{m-1}} Pr(\pi_{m-1} \oplus x_{k-1}^k | \pi_{k-1} \text{ ranks the top-}(k-1) \text{ among S})$$

$$= c$$

By Lemma 3 and induction hypothesis, we get $Pr(\pi_k | S) = q^{d_S(\pi_k) + L_S(\pi_k)} \cdot \frac{\psi(|S| - k, q)}{\psi(|S|, q)}$. Hence our statement holds for $k = K$, too, thus finishing the proof. $\square$

**Proof of Theorem 5.** In this proof, we use the $\sigma$ notation for a ranking in order to simplify the algebra. Denote the parameter as $\theta = (\sigma, q)$ and $Pr_\theta(\cdot)$ is the choice probability under parameter $\theta$. The notations $d_S^\sigma$ and $L_S^\sigma$ are the original ones $d_S^\sigma$ and $L_S^\sigma$ under the ranking parameter $\sigma$, respectively.

For all voting history $H_T = (S_1, \pi_k^1, \ldots, S_T, \pi_k^T)$, define $\pi_k^t$ as $[x_1^t, \ldots, x_k^t]$. Denote the central ranking as $\sigma$, then the log likelihood is

$$\sum_{t=1}^T \log Pr_\theta(\pi_k^t \mid S_t) = \sum_{t=1}^T \left[ \log \frac{\psi(|S_t| - k, q)}{\psi(|S_t|, q)} + \log q \cdot (d_{S_t}^\sigma(\pi_k^t) + L_{S_t}^\sigma(\pi_k^t)) \right]$$

$$= \sum_{t=1}^T \sum_{i=|S_t|-k+1}^{|S_t|} \log \frac{1-q}{1-q^i} + \log q \cdot \left( \sum_{t=1}^T d_{S_t}^\sigma(\pi_k^t) + L_{S_t}^\sigma(\pi_k^t) \right) \quad (*)$$

By definition, we can get

$$(*) = \sum_{t=1}^T \sum_{i=|S_t|-k+1}^{|S_t|} \log \frac{1-q}{1-q^i} + \log q \cdot \sum_{(i,j):i \neq j} \mathbb{I}\{\sigma(j) < \sigma(i)\} \cdot \left( \sum_{t=1}^T \mathbb{I}\{x_k^t = i\} \cdot \mathbb{I}\{i, j \in S_t \backslash \{x_1^t, \ldots, x_{k-1}^t\}\} + \right.$$

$$\left. (|S_t| - 1) \cdot \mathbb{I}\{x_1^t = i, x_2^t = j\} + (|S_t| - 2) \cdot \mathbb{I}\{x_2^t = i, x_3^t = j\} + \ldots + (|S_t| - k + 1) \cdot \mathbb{I}\{x_{k-1}^t = i, x_k^t = j\} \right)$$

$$= \sum_{t=1}^T \sum_{i=|S_t|-k+1}^{|S_t|} \log \frac{1-q}{1-q^i} + \log q \cdot \sum_{(i,j):i \neq j} \mathbb{I}\{\sigma(j) < \sigma(i)\} \cdot \left( \sum_{t=1}^T \left[ \mathbb{I}\{x_k^t = i\} \cdot \mathbb{I}\{i, j \in S_t \backslash \{x_1^t, \ldots, x_{k-1}^t\}\} + \right. \right.$$

$$\left. \left. \sum_{h=1}^{k-1} (|S_t| - h) \cdot \mathbb{I}\{x_h^t = i, x_{h+1}^t = j\} \right] \right)$$

Hence, in the integer program (5), we define $w_{ij}$ as follows.

$$w_{ij} := \sum_{t=1}^T \left[ \mathbb{I}\{x_k^t = i\} \cdot \mathbb{I}\{i, j \in S_t \backslash \{x_1^t, \ldots, x_{k-1}^t\}\} + \sum_{h=1}^{k-1} (|S_t| - h) \cdot \mathbb{I}\{x_h^t = i, x_{h+1}^t = j\} \right].$$

That finishes the proof. $\square$

# E  The Expectation-Maximization (EM) for Mixture OAM

In this section, we describe the standard expectation maximization (EM) algorithm used to fit a mixture of $K$ OAMs to a given sample of choice data. This particular version is based on [4] and adapted for the OAM setting.

First, introduce the following notation.

- $K$: number of clusters.
- $Z_t : (c_{t1}, ..., c_{tK})$, $c_{tk} \in \{0, 1\}$ indicates whether sample $t$ comes from cluster $k$.
- $\Theta = (\{p_k\}, \{\sigma_k\}, \{\alpha_k\})$: $p_k$: mixture probability of cluster $k$, $\sigma_k$ : central ranking of cluster $k$, $\alpha_k$: concentration parameter of cluster $k$.
- $OAM_k(x|S)$: choice probability of item $x$ given display set $S$ under cluster $k$.

To fit the mixture distribution, we would like to solve the following maximum likelihood problem:

$$\max_{p,\sigma,\alpha} \sum_{k=1}^{K} \sum_{t=1}^{T} c_{tk} \log p_k + \sum_{k=1}^{K} \sum_{t=1}^{T} c_{tk} \log[OAM_k(x_t|S_t)]$$

The EM algorithm starts with an initial solution and iteratively obtains an improving solution until an appropriate stopping criterion is met. Suppose $\{(p_k, \sigma_k, \alpha_k) : 1 \leq k \leq K\}$ is the current solution. Then, an improving solution $\{(\hat{p}_k, \hat{\sigma}_k, \hat{\alpha}_k) : 1 \leq k \leq K\}$ is obtained as follows.

In the Expectation step, the algorithm computes "soft counts" $c_{tk}$, denoting the (posterior) probability that example $t$ was generated from mixture component $k$. This probability is computed as

$$\hat{c}_{tk} = \frac{\hat{p}_k * O\hat{A}M_k(x_t|S_t)}{\sum_{j=1}^{K} \hat{p}_j * O\hat{A}M_j(x_t|S_t)}$$

Then, in the Maximization step, we first set $\hat{p}_k = \frac{1}{T} \sum_{t=1}^{T} \hat{c}_{tk}$, and then solve a separate optimization problem for each segment $1 \leq k \leq K$ :

$$\max_{\sigma_k, \alpha_k} -\alpha_k \sum_{t=1}^{T} c_{tk}(\sigma_{S_t}(x_t) - 1) - \sum_{t=1}^{T} c_{tk} \log \sum_{j=0}^{|S_t|-1} e^{-\alpha_k j}.$$

We can obtain an optimal solution of the above problem by solving the following two problems:

$$d_k^* = \min_{\sigma_k} \sum_{t=1}^{T} \hat{c}_{tk} \cdot (\sigma_{S_t}(x_t) - 1) \tag{16}$$

$$\hat{\alpha}_k = \arg\min_{\alpha_k} \left\{ \alpha_k \cdot d_k^* + \sum_{t=1}^{T} \hat{c}_{tk} \log \sum_{j=0}^{|S_t|-1} e^{-\alpha_k j} \right\} \tag{17}$$

The problem in (16) can be solved by choice aggregation through the integer program (5) (with properly defined $w$ scores). Also, the optimization problem in (17) is convex.

Once we obtain the new solution, $\{(\hat{p}_k, \hat{\sigma}_k, \hat{\alpha}_k) : 1 \leq k \leq K\}$, the above process is repeated until the increase in the log-likelihood value falls below a certain threshold, or until a prespecified limit on the number of iterations is reached.

# F    More Experiments

**Experiment 1.1: More analysis on the results of experiment 1.** In experiment 1 of section 4, We randomly split the data into 4000 preference rankings in training and 1000 preference rankings in testing. When we train mixed-MNL on the full set, the mixed-MNL with only one cluster can already perfectly explain the choice data. That is, the predicted choice probabilities are exactly equal to the empirical choice probabilities (aka the market shares). Hence, the learned parameters of mixed-MNL with a smaller number of clusters are also optimal solutions to the mixed-MNL with a larger number of clusters. Specifically, the optimal parameters of $i$-cluster MNL , $i \in \{1, 2, \ldots, k\}$ are all optimal to $k$-cluster MNL. In other words, mixed-MNL with multiple clusters has many optimal solutions.

These optimal parameters can predict the same choice probabilities on the full set but may predict differently on other sets. Given the number of clusters, we record the best log likelihood value and worst log likelihood value from the multiple optimal parameters. We repeat the random splitting 3 times and show these 3 results in the Figure 2.

Figure 2: Train on the full display set only (Each row is on a random data splitting.)

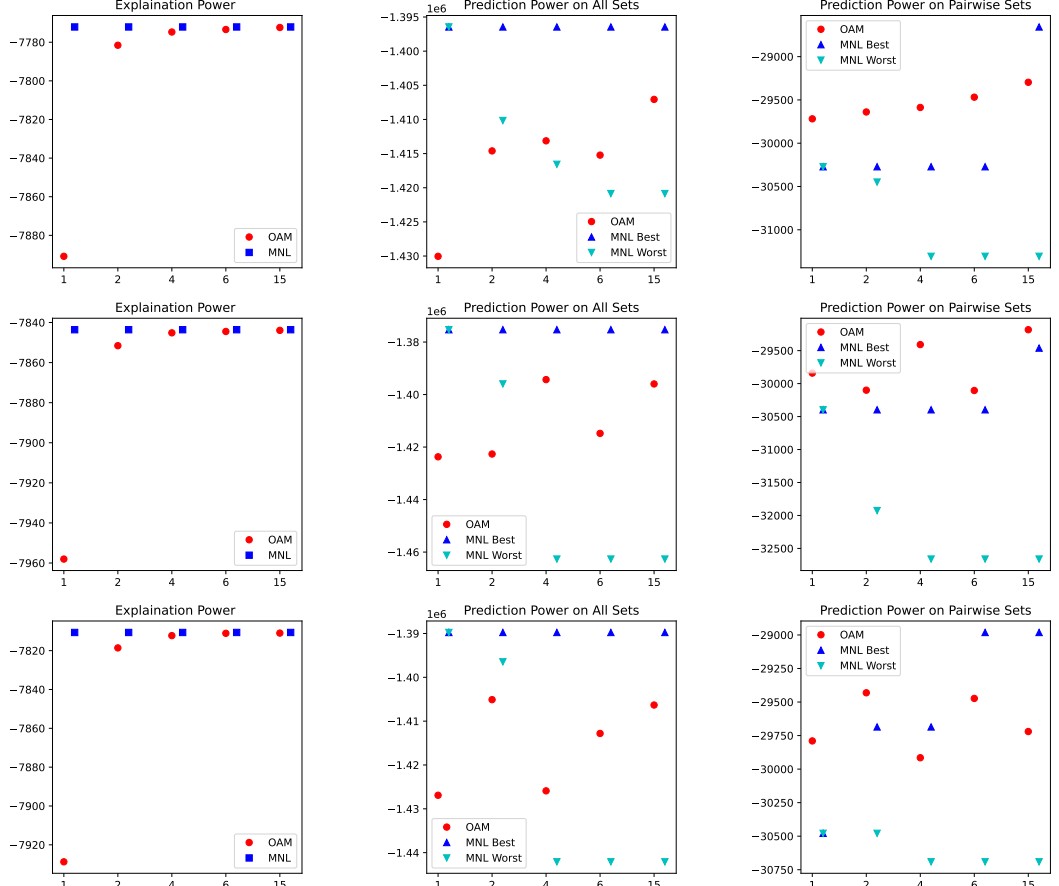

In each panel, the x-axis represents the number of clusters in the mixture model, and the y-axis represents the log-likelihood metric.

In these three figures, fix the number of clusters, the prediction power of mixed-MNL may span wide ranges between the best record and the worst record.[6] However, the prediction power of OAM are in or beyond the range across different number of clusters under two test setups.

**Experiment 1.2: Repeatedly shuffle the data.** Now, we repeat the data splitting 80 times to obtain multiple performance values.

We use $\bar{ll}$ to denote the mean of log-likelihood values from the 80 times data splitting and $std(ll)$ to denote the standard deviation of log-likelihood values. The confidence interval (CI) is defined as $[\bar{ll} - 1.96 \cdot \frac{std(ll)}{\sqrt{SS}}, \bar{ll} + 1.96 \cdot \frac{std(ll)}{\sqrt{SS}}]$, where the sample size $SS = 80$.

The result of training on the full set is in Figure 3. Given the collection of random sets in experiment 1 of section 4, the result is in Figure 4. From these two figures, the prediction power of OAM increases with the number of clusters increases. Hence, OAM seems to suffer less from overfitting.

**Experiment 2.1: Compare with Borda Count and Simple Count.** Since classical Borda Count could only be applied on the full set, we adapt it as follows: If a display set contains $M$ items and there is a top-$k$ ranked choice, the ranked items in the ranked choice get $(M-1), (M-2), \ldots, (M-k)$

---

[6]Since we only try out a limited number of optimal solutions, the actual range could be much wider.

Figure 3: Train on full set cross shuffle

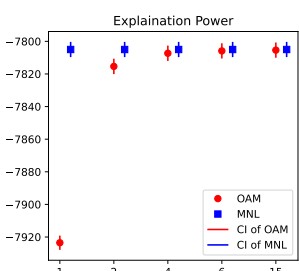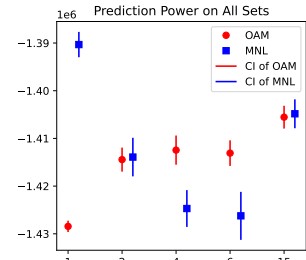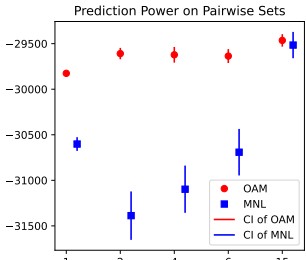

Figure 4: Train on three random display sets cross shuffle

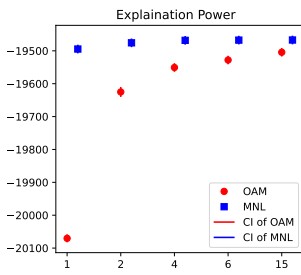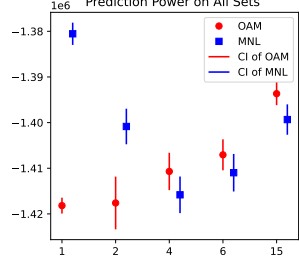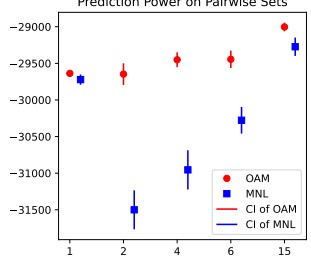

scores. We also consider an even simpler variation of the Borda count where the scores are unweighted. Because of the simplicity of this variation, we refer to it as Simple Count.

First, we compare three methods on the setups used in experiment 2 of section 4, where all sets with sizes larger or equal to $k$ will be displayed when the feedback is top-$k$ ranked choice. The learned rankings under top-1, top-2, top-3 feedback structure are shown in Table 2.

|  | OAM | Borda Count | Simple Count |
|---|---|---|---|
| Top-3 | (8,5,6,3,2,1,4,9,7,10) | (8,3,6,1,2,5,4,9,7,10) | (8,3,1,6,2,5,4,9,7,10) |
| Top-2 | (8,5,6,3,2,1,4,9,7,10) | (8,3,5,6,1,2,4,9,7,10) | (8,3,6,1,5,2,4,9,7,10) |
| Top-1 | (8,5,6,3,2,1,4,9,7,10) | (8,5,6,3,2,1,4,9,7,10) | (8,5,3,6,2,1,4,9,7,10) |

Table 2: Learned rankings under different feedback structure when display sets are balanced.

To measure the discrepancy among the estimated rankings from top-1, top-2, and top-3 feedback, we use the average pairwise Kendall's Tau distance among the estimated rankings. The results are $0, 4$, and $4$, respectively, for OAM, Borda Count, and Simple Count, respectively. As a result, we can see that OAM is the most stable one.

Furthermore, and perhaps more importantly, Borda Count and Simple Count become particularly ineffective when the display sets are unbalanced because those simple metrics fail to incorporate the display set information. For example, if the learner disproportionally displays the least preferred items (items 7,9,10 in our data), those items will be ranked high under Borda/simple counts because only the "counts" matter. In comparison, the RMJ-based method judiciously adjusts the count of an item by the display set history.

In Table 3, we show the learned rankings under top-1, top-2, top-3 feedback structure when only the full set $[10]$ and $\{7, 9, 10\}$ are displayed. The average pairwise Kendall's Tau distance values of these three methods are $2.67, 7.33$ and $6$. OAM is still the most stable one under this unbalanced setting. In addition, from Table 3, OAM still ranks item $7, 9, 10$ at the bottom, while Borda Count puts items $7, 9$ at some high positions and Simple Count put items $7, 9, 10$ at very top positions.

Let us summarize our findings in Experiment 2.1 and the advantage of OAM compared to different variants of the Borda count. First, under both balanced and unbalanced settings, OAM is the most

| | OAM | Borda Count | Simple Count |
|---|---|---|---|
| Top-3 | (8,5,3,2,6,1,4,9,7,10) | (8,3,5,6,9,2,1,7,4,10) | (9,7,10,8,3,5,6,2,1,4) |
| Top-2 | (8,5,6,3,2,1,4,9,7,10) | (8,5,9,6,3,2,1,7,4,10) | (9,7,8,10,5,6,3,2,1,4) |
| Top-1 | (8,5,2,6,1,3,4,9,7,10) | (8,5,9,7,2,6,1,3,4,10) | (9,7,8,5,2,6,10,1,3,4) |

Table 3: Learned rankings under different feedback structures when display sets are unbalanced.

stable method. Second, Borda Count and Simple Count will learn misleading ranking when the display sets are unbalanced, i.e., some items are displayed much more times than other items, while OAM won't be affected by this unbalance. Finally, it is worth pointing out that Borda Count and Simple Count are only about the central ranking, while OAM is also about making (probabilistic) predictions on ranked choices.

**Discussion: further comments on the numerical studies.** We believe that our numerical experiments have demonstrated promising evidence that the RMJ-based ranking model can be used to capture people's ranked choices from varied display sets. As such, it adds to the toolbox for our motivating application in preference learning in contexts such as crowdsourcing, marketing research, and survey designs. Since our model is a generalization of choice modeling (i.e., every participant makes a single choice out of a subset of items), we believe it can find potential applications in other domains such as online/brick-and-mortar retailing. Of course, more comprehensive numerical experiments are called for to better evaluate its potential, which we leave for future research. For example, we could use data sets that specialize in the retail setting. In addition, it would be interesting to see how the current method handles the non-purchase option since it may well be the overwhelmingly most frequent response in the data. Finally, it would be beneficial to explore how the current method can be extended to incorporate customer covariates.

**Experiment 4: Experiment on the YOOCHOOSE dataset for e-commerce.** We wish to conclude this section by showing some initial sign of success in the e-commerce context. We compare our model with MNL on the YOOCHOOSE dataset of the RECSYS 2015 challenge, which contains six months of user activities for a large European e-commerce business [23].

Here is a brief description of the data set and the data cleaning rule. In this data set, click and purchase data are provided at the session level. In every session, we can observe the collection of products that the customer clicks and the collection of products that the customer buys. We use the click data to form the "display sets" and purchase data to form the "choices" in our paper.[7] We conduct our experiment on category 12. In particular, we focus on the top 111 items, which consist of over $95\%$ click data. If there are purchases of multiple products in one session, we randomly choose one item as the customer's choice. If the customer buys more than one same item in a single session, we also treat it as a single choice. In the end, there are 76 items and 274 choice data points after data cleaning.

Here is a brief description of how to randomly shuffle and split the data into training and testing sets. To ensure that all items appear in the training set, we keep 53 choice data in the training set and randomly split the other 221 choice data into the training and testing set. Finally, there are 220 choice data in the training set and 54 choice data in testing. We randomly repeat the data splitting 100 times. On average, there are 124.75 unique display sets in each training set and 2.72 items in a display set. Note that this is an extremely sparse collection, considering the power set of 76 items consists of $2^{76} \approx 7.56 \times 10^{22}$ elements.

In this experiment, we use one cluster OAM and MNL. The results are shown in Figure 5. From this figure, MNL achieves better explanation power, but OAM performs better prediction power on average. This is a sign that MNL tends to overfit in this data set where the coverage of the display set is limited. In comparison, OAM demonstrates better generalization ability.

# G Code and Data

Our code and data can be accessed by this link: https://drive.google.com/drive/folders/1q-NUkX21IaqFVXvbLsgVByzXYIR3oqOH?usp=sharing.

---

[7]We find that the no-purchase option is overwhelmingly frequent in the data. To deal with this issue and compare the customer preferences over the real items, we filter out the non-purchase and single-click sessions.

Figure 5: Results of 100 times data splitting on the YOUCHOOSE dataset

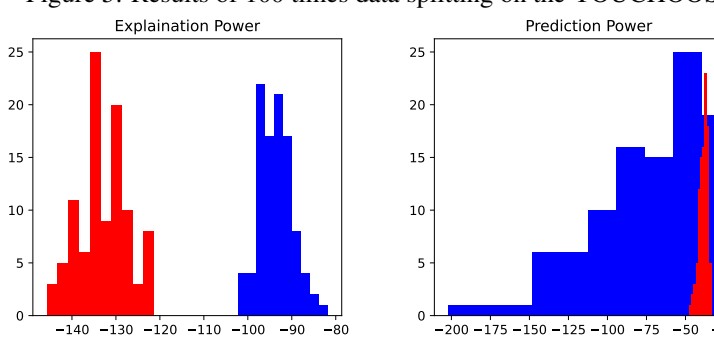

In each panel, the x-axis represents the log-likelihood metric and the y-axis represents the frequency.