# OpenReview forum: "On A Mallows-type Model For (Ranked) Choices"
_NeurIPS.cc/2022/Conference — NeurIPS 2022 Accept_

### Official Review · Reviewer_Gwow · 2022-07-10

**Rating:** 6
**Confidence:** 4
**Soundness:** 3 good
**Presentation:** 3 good
**Contribution:** 3 good

**Summary:**

This paper discusses a new ranking model called Reverse Major Index, which is a friend of the classical Mallows ranking model. Various properties of the model including the MLE are discussed, and connection to integer programming is discussed.

**Questions:**

The paper is well written and I don’t have question.

**Strengths And Weaknesses:**

The paper is well written and scientifically correct. The model discussed in the paper are novel up to my knowledge.
However, the model seems to me close to the Mallows, and the analysis seems to be similar. The authors should emphasize more on why the benefit of this ranking model.

---

> ### Author Response · Authors · 2022-08-02
> **Thanks and responses from authors!**
>
> We thank you for your encouraging comments. Please see our responses below.
>
>
> The main benefit of the new ranking model is that it provides a tangible tool to estimate a (mixture of) distance-based ranking model from people's ranked choices from varied display sets. Other ranking-model-based methods are either prohibitively time-consuming (e.g., MLE for Mallows model), or suffer from provable consistency issues (e.g., Mallows Smoothing for Mallows model), or suffers from overfitting issues (e.g, data-driven methods with sparse distributions over rankings).

---

### Official Review · Reviewer_L5sw · 2022-07-10

**Rating:** 7
**Confidence:** 4
**Soundness:** 3 good
**Presentation:** 3 good
**Contribution:** 3 good

**Summary:**

The authors study Mallows-type models in the context of top-$k$ ranking.
They propose a new distance function called the reverse major index known from combinatorics,
with which the computation of choice probabilities and the sampling of rankings can be done efficiently.
Parameter inference reduces to the feedback arc set problem, which is hard but admits a polynomial-time approximation scheme.
The authors show that their maximum likelihood estimator is consistent, while this does not hold for the Mallows smoothing heuristic from the literature.
They derive closed-form equations for the computation of the choice probabilities of their model.

To validate the model, experiments on the Sushi datasets are performed. In the case of top-1 choice, the approach shows favorable
performance when compared to Mallows smoothing and the multinomial logit model.

**Questions:**

- In line 330f. you write that your method has "supreme performance compared to MS and MNL across the settings".
  Looking at Figure 1, the prediction power of OAM and MNL on all sets are actually close, with MNL beating OAM for a few cluster configurations.
  Since there are no error bars/confidence intervals, it is unclear as to how significant this outperformance really is.
  Can you clarify
      (1) why it would not be possible to repeatedly shuffle the data and repeat the splitting to obtain multiple performance values, such that error bars can be computed?
      (2) why "MNL is vulnerable to [...] overfitting" is not applicable to OAM as well, since the prediction power of OAM appears to decrease similarly for more clusters.
- Could you elaborate a bit what the goal of Experiment 2 was? What ranking do other top-k ranking approaches produce on this task? Which ranking do you get if you simply count the occurences of each sushi in the top-k list?
- Similarly, the goal of Experiment 3 was also not explained. One can deduce that you are trying to gauge how well the method can scale to large choice sets. Are there other approaches you can compare to which will help contextualize your obtained running times?

**Limitations:**

The authors write about the potential limitations of their model at the end of Section 3. What I am missing, are the limitations of their numerical experiments. The authors could for example write about the choice of dataset(s), baselines, number of repetitions etc., which in addition to more motivation would help the reader contextualize the results.

**Strengths And Weaknesses:**

# Originality
The authors propose a novel combination of the Mallows model with a new distance function, the *reverse major index*.
The distance function only considers neighboring ranks and the absolute position in the ranking, which makes it easier to handle than Kendall’s distance function.
Subsequently, this allows the authors to simplify computation of several downstream tasks like sampling and inference, and allows them to show consistency of their model.

Immediate related work is adequately cited, and the authors are referencing some of the choice modeling literature.
Since the setting considered in the paper is dealing with a feedback model where users are ranking a given set of items, I would also recommend specifically citing the recent work done in the setting of subset choice:

- Benson, Austin R., Ravi Kumar, and Andrew Tomkins. “A Discrete Choice Model for Subset Selection.” In Proceedings of the Eleventh ACM International Conference on Web Search and Data Mining, 37–45. WSDM ’18. New York, NY, USA: Association for Computing Machinery, 2018. https://doi.org/10.1145/3159652.3159702.
- Pfannschmidt, Karlson, Pritha Gupta, Björn Haddenhorst, and Eyke Hüllermeier. “Learning Context-Dependent Choice Functions.” International Journal of Approximate Reasoning 140 (January 1, 2022): 116–55. https://doi.org/10/gn9kh3.

# Quality

## Theoretical analysis
The authors prove several theoretical results related to the computation of probabilities, sampling and inference.
In addition, they show that the maximum likelihood estimator of their RMJ model is consistent, while Mallows smoothing fails in this respect.
Notation is clearly introduced and used consistently. The proofs of the claims are provided in the supplemental material. The proofs themselves are written clearly, such that they were straightforward to follow.

In the proof of Lemma 4, the proof of the identifiability is very short and could benefit from slightly more intermediate steps.
In line 560, you refer to a source [20], but in the paper the references only go up to [19]. It is therefore not clear where these conditions come from.

## Empirical analysis
While the theoretical analysis is sound and thorough, the numerical experiments are more limited in scope.
The authors only consider the Sushi dataset, which is unique in that it is one of the rare datasets where users provide rankings for the chosen items. Since the authors then consider top-$1$ choice, it would be possible to consider more real-world data (see below), and more baselines of the choice modelling literature. As is, it is difficult to relate the results to the results of the existing literature.
The most interesting experiments (2 and 3) could be expanded by comparing the results to simple baselines (see Questions section) or adapting existing ranking/choice approaches to this setting.

Examples for real-world datasets:

- The YOUCHOOSE dataset of the RECSYS 2015 challenge: https://web.archive.org/web/20210120062757/https://2015.recsyschallenge.com/challenge.html
- The Expedia hotel recommendation dataset: https://www.kaggle.com/competitions/expedia-hotel-recommendations/data
- The LETOR datasets: https://www.microsoft.com/en-us/research/project/letor-learning-rank-information-retrieval/

# Clarity
The paper is written in a clear language and structured logically. Titles and paragraphs are used in such a way that it is easy to follow the key ideas. The source code and data of the experiments is provided with a small readme file, which improves the reproducibility of the results (disclaimer: I did not execute the attached code).

In the experimental section, since the generation of some of the datasets has been randomized, I am missing a description of whether random seeds were fixed beforehand and to which value. Looking at the attached code, this appears not to be the case, but it should be mentioned.
# Significance
The model proposed in the paper improves on the standard Mallows model, by making it more efficient to learn and use. The authors also demonstrate that the approach scales to moderately sized data (i.e., choices from sets of 100).
Whether others will use the approach will depend on the predictive performance achieved on real-world data, when compared to other approaches.
I think that the idea is interesting as is and could lead to other advances in the subsequent literature.

# Minor remarks

- Line 1: Preference learning is a much more general field in machine learning and not only encompasses top-k choice/ranking.
- L. 42: "A representative work is by [4]" -> "A representative work is by Antoine et al. [4]"
- L. 320: "traing" -> "training"
- L. 599: "we will show that even [when] all display sets"

---

> ### Author Response · Authors · 2022-08-02
> **Thanks and responses from authors!**
>
> We thank you very much for your helpful comments and your positive tone. Please see our responses below.
>
> [Theoretical analysis]
>
> We have completely rewritten the proof of the identifiability (i.e., Theorem 2) with significantly more intermediate steps; see Appendix B in the rebuttal revision. The original intended source [20] was "Thomas S. Ferguson. A course in large sample theory. London: Chapman & Hall, 1996," which was a textbook reference. The new proof is improved: we now exploit the problem structure and directly prove the uniform convergence of the likelihood function, rather than invoking standard (sufficient) conditions. Therefore,  [20] is not needed anymore.
>
> [Empirical analysis]
>
> We agree that there is room for a more comprehensive empirical study. Since our main innovations are on the theoretical side and there was a tight page limit, we were only able to include a limited collection of numerical experiments. We have now included more numerical studies in the supplementary material of the rebuttal revision, which we will discuss further below.
>
> Thank you also for providing the alternative datasets. We believe it important to try contexts beyond the sushi preference data and tried very hard to include those. But unfortunately, we did not have enough time to clean the data and re-conduct the analysis before the rebuttal deadline. Therefore, we leave that as an open question.
>
>
>
> [Clarity]
> Previously the random seed was set as the default value. Now we have revised the code to fix and report the seeds easily.
>
>
> [Significance]
> Thank you for your encouraging comments.
>
> [Questions]
>
> Q1: Your comments inspired us to take a much closer look at how OAM compares with MNL (since the advantage of OAM over MS seems pretty clear); please see Appendix F in the rebuttal revision. Given the same number of clusters, MNL is more likely to overfit than OAM. The reason is that every cluster carries not only the "ordinal" information but also the "cardinal" one. We believe the point is best made in the extreme case where only the full display set is included: In this case, a single-cluster MNL is sufficient to perfectly match the empirical choice probabilities (i.e., market shares). A single cluster OAM cannot do the same because of the fixed exponentially decaying shape of choice probabilities.
>
> In our numerical experiments, we confirm this insight in Figure 2: Unlike OAM, the single-cluster MNL already perfectly explains the market shares, and the explanation power stops improving after that. When it comes to prediction power, the MLEs are typically not unique. (For example, if one cluster already perfectly explains the market shares, the one-cluster solution is also optimal under the multi-cluster setting.) We find that the performance of MNL is not necessarily all bad. But it **highly** depends on the tie-breaking rule: an uncarefully chosen MLE (since they are non-unique) can perform significantly worse than OAM. In comparison, OAM seems to suffer less from overfitting as its prediction power increases steadily as the number of clusters increases.
>
> We are also able to include data shuffling. We find that our insight is robust to random splitting of training/testing datasets; see Figures 3 and 4.
>
>
>
> Q2: Experiment 2 is a robustness check based on the criterion that a good model should return similar central rankings under different feedback structures, i.e., different $k$. The result presents evidence that our methodology is learning sensible information from real-world data.
>
> We also conduct additional studies on other top-k ranking approaches; see Appendix F. For example, one could use the Borda count equivalent or simple counts. We find that both of them are ineffective since they fail to incorporate the display set information. (For example, if the learner disproportionally displays the worst two items, those items will be ranked high under Borda/simple counts. The RMJ-based method judiciously adjusts the count of an item by the display set history.) The Mallows and Plackett-Luce models would also be decent candidates, but we do not know of any tractable way of estimating those models based on ranked choices from varying display sets.
>
> Q3: The goal of experiment 3 is to show that our method is still effective when there are a relatively large number of items. The best apple-to-apple comparison with our method would be MLE for the Mallows model. However, regarding the MLE problem for Mallows, we do not know any method significantly better than the brute force method, whose time complexity is $\Omega (n!)$. That makes the implementation of MLE for Mallows prohibitively difficult.
>
> [Other comments]
> Thank you for pointing out additional references. We find them relevant and plan to cite them in the camera-ready version. Also, we will clarify the limitation of numerical experiments (e.g., ability to handle customer covariates, highly unbalanced data, etc.)

---

> > ### Comment · Reviewer_L5sw · 2022-08-03
> > **Revised Theorem 2**
> >
> > I checked the revised version of Theorem 2 and the proof thereof. The proof of the "if" part is very straightforward. Regarding the "only if" part I have a question:
> > Why is it possible to say *without loss of generality* that $\\{n-1, n\\}$ is not covered at all. Could it not be the case that you observe a pair $\\{i, j\\}$ finitely often such that $(\hat{\pi}, \hat{q}) \to (\pi^*, q^*)$? For such a pair, no tie-breaking rule would exist. Let me know, if I misunderstood something.
> >
> > Minor Typo in Appendix line 569: "tie-breaknig" -> "tie-breaking"

---

> > > ### Author Response · Authors · 2022-08-06
> > > **Responses of comment on the revised Theorem 2.**
> > >
> > > We greatly appreciate your careful read of the proof of Theorem 2 and your further comments. Below are our responses.
> > >
> > > We have further revised the proof of Theorem 2 with more details provided in the "only if" part. (In particular, we write the new proof not assuming that "\{n-1,n\} is not covered at all without loss of generality" because we realize that it can cause confusion.)
> > >
> > > -  Coming back to your question, it turns out if one observes a pair \{i,j\} only finitely often, it is impossible to guarantee consistency. Why so? Let us provide a short answer here. If \{n-1,n\} is only covered finitely many times, both the events $\{w_{n-1, n} \geq w_{n, n-1}\}$ and $\{w_{n-1, n} \leq w_{n, n-1}\}$ happen with positive probability in the limit. That makes MLE unable to tell the rankings of items $(n-1)$ and $n$ (even in the limit) if they are bottom-ranked items. The complete proof is contained in Appendix B.1 of the further revised paper.
> > >
> > > Finally, we thank you for catching the typo and have corrected it in our revision.

---

> > > > ### Comment · Reviewer_L5sw · 2022-08-09
> > > > **Revised proof of Theorem 2**
> > > >
> > > > I checked the new proof for Theorem 2, and it is now straightforward to follow the argument. What I was missing is that you are not talking about both events happening with positive probability during the same sampling trajectory, but a priori. So with some constant probability, however small, the bad event can happen and prevent consistency.
> > > >
> > > > One additional typo I found:
> > > > Theorem 2 (line 243): "is display[ed] infinite[ly] often"

---

> > > > > ### Author Response · Authors · 2022-08-09
> > > > > **Thank you.**
> > > > >
> > > > > We thank you for your additional efforts in reading the proof of Theorem 2 and your help in clarifying the results and the proof.

---

> > ### Comment · Reviewer_L5sw · 2022-08-03
> > **Additional experiments & clarifications**
> >
> > First of all, thank you a lot for the additional experiments you performed in this short timeframe. They are much appreciated.
> > I regard the model and your theoretical work to be your main contributions. As such, I do not expect you to have comprehensive results on real-world data as well. The mentioned datasets were just pointers for you to consider in the future.
> > You should therefore focus on motivating the existing experiments well, and presenting the key insights clearly, while avoiding exaggeration like "supreme performance".
> >
> > Regarding Experiment 1.1: Since you added data shuffling for your experiments in the appendix, are you also planning on repeating Experiment 1 using shuffling such that you can include error bars in Figure 1?
> >
> > Experiment 2.1 in my view gives the audience a much better idea of the robustness of OAM than what is currently reported in experiment 2.
> > I know that the page limit is very restrictive, but maybe some of the insights can be moved to the main paper.
> >
> > Regarding your answers to Q2 and Q3: These are good descriptions of the experiments, and you should include those in the main paper to motivate the experiments to the reader.

---

> > > ### Author Response · Authors · 2022-08-06
> > > **Responses of comments on additional experiments & clarifications.**
> > >
> > > We greatly appreciate your careful read of the additional experiments and your further comments. Below are our responses.
> > >
> > > We have revised the "numerical experiments" section in the main paper as well as the "more experiments" section in the appendix with a focus on motivating the existing experiments and presenting the key insights. In particular, we have fleshed out the descriptions of the experiments in the main paper to motivate the experiments. Regarding your comment about the language, thank you and by no means did we intend to overstate the results. We have revised the language to reflect our original intention more authentically.
> > >
> > > Regarding the robustness against data shuffling: Our Experiment 1.2 in Appendix F (formally experiment 1.1) is to serve this purpose. In particular, given that the advantage of OAM over Mallows Smoothing seems clear, we believe Figures 3 and 4 in Appendix F capture what you asked for. To this end, we wish to also clarify that we have identified two sources of performance variation: the randomness of training-testing data split and non-uniqueness of solution (due to identifiability issues). We find that the latter dominates the former. Since we find it difficult to visualize both effects in a single plot, we have included the additional numerical experiments in the appendix. But we have included a pointer to those additional experiments.
> > >
> > > Regarding Experiment 2.1: Thank you for your encouraging comments. We have added insights to the main paper up to what the page limit allows. We also have included a pointer to this additional experiment.
> > >
> > >
> > > Finally, we have some additional updates from your earlier comments. We have clarified the limitation of numerical experiments in the further revised paper; see around line 809. Also, we are happy to share that we are able to obtain results from the alternative data sets you suggested earlier. (We started the experiments during the rebuttal period, but the experiments only finished during the discussion period.) We looked at the YOOCHOOSE e-commerce dataset since we feel it is most relevant to our setting. We find that OAM continues to show favorable results compared to MNL. Perhaps more importantly, the intuition remains robust: since the variation of the display set is highly limited, MNL seems to suffer from overfitting issues, which explains why OAM demonstrates better generalization power. More details of the numerical study are contained in Experiment 4 of Appendix F.

---

> > > > ### Comment · Reviewer_L5sw · 2022-08-09
> > > > **Revised experimental section**
> > > >
> > > > I read the revised version of the experimental section and I agree that it is now much clearer what you are doing and why.
> > > >
> > > > The results on the YOUCHOOSE dataset are certainly favorable, and I commend you adding these results in such a short time frame.
> > > >
> > > > Typo in Line 351: "Border count" -> "Borda count"

---

> > > > > ### Author Response · Authors · 2022-08-09
> > > > > **Thank you.**
> > > > >
> > > > > We thank you for your additional efforts in reading the experimental section and encouraging comments.

---

> > ### Author Response · Authors · 2022-08-09
> > **Thank you!**
> >
> > Thanks again for reviewing our paper. Your thoughtful comments have undoubtedly helped strengthen the paper, and we really enjoy communicating with you. We hope your questions or concerns have been fully addressed. In case you have any additional questions, we are also prepared to answer them. Thank you very much for your attention.

---

> > > ### Comment · Reviewer_L5sw · 2022-08-09
> > > **Adjusted rating**
> > >
> > > After the fruitful discussions with the authors of the paper, my questions have been addressed. The authors revised the proof of Theorem 2, added new experiments (including new real-world data) and improved the description of the results.
> > >
> > > Therefore, I increased my overall rating from 6 to 7 (Accept).

---

> > > > ### Author Response · Authors · 2022-08-09
> > > > **Thank you!**
> > > >
> > > > Once again, we sincerely thank you for your efforts and help in making our paper stronger (with multiple aspects ranging from theory to numerical studies and to writing). That is truly valuable to us!

---

### Official Review · Reviewer_nVtJ · 2022-07-10

**Rating:** 7
**Confidence:** 4
**Soundness:** 3 good
**Presentation:** 4 excellent
**Contribution:** 3 good

**Summary:**

This paper proposed a Mallows-type (probabilistic) model for ranking data. Based on the new model, several theorems/lemmas are proved to  show different properties of the new model. Using the properties, the authors proposed a learning algorithm for the new model, and experimentally showed that the new model has better accuracy than the widely-used Plackett-Luce model and Mallows model.

**Questions:**

As mentioned in the weakness part, can the authors provides more discussions about the time-complexity?

Also, Please correct me if my understanding of the distance measure is incorrect.

Why is linear weight used in the distance measure? Why not use other weights like log-based weighting in DCG?

**Limitations:**

I haven't found any negative social impact in this paper

**Strengths And Weaknesses:**

Strengths:

1. The presentation is very clear! It's easy for me to find the main contribution of this highly theoretical paper

2. The theorems look sound to me

3. The idea is simple. It looks similar to the intuition behind DCG (or NDCG).

Weakness:

1.  I think the model is not novel because it can be seen as a weighted version of Mallow's model with Cayley distance (Irurozki, Ekhine, Borja Calvo, and Jose A. Lozano. "Sampling and learning Mallows and Generalized Mallows models under the Cayley distance." Methodology and Computing in Applied Probability 20.1 (2018): 1-35.). If my understanding is correct, the baseline should also be changed to the Mallow's with Cayley distance. Please correct me if the proposed measure is totally different from Cayley distance. PS: My current (overall) evaluation is based on the assumption that the proposed measure is similar to Cayley distance.

2. I would  like to see more discussions about the time complexity of the new algorithm. I think it would be a big improvement if the authors can provide the analysis about the overall time-complexity of this algorithm in Section 3.1. Or, alternatively, it would also be great if the authors can show a comparison of Mallows and OAM in terms of the running time.

---

> ### Author Response · Authors · 2022-08-02
> **Thanks and responses from authors!**
>
> We thank you very much for your helpful questions and comments. Please see our responses below.
>
> Q1 (Time complexity)
>
> The RMJ-based ranking model aggregates into the OAM choice model. Therefore, the MLE problem for the central ranking $\pi^\ast$ reduces to the **feedback arc set problem on tournaments** (FAST). While FAST is NP-hard, it admits a PTAS (Kenyon-Mathieu, Claire, and Warren Schudy. "How to rank with few errors." Proceedings of the thirty-ninth annual ACM symposium on Theory of computing. 2007.) That is, for any fixed error $ \epsilon $, there is a polynomial-time algorithm that achieves $(1-\epsilon)$ optimality of our MLE problem. Besides PTAS, integer-programming-based methods are also (if not more) useful in practice when the data is moderately sized and generated either randomly or in the real world; please see Feng et al. (2021) for a more detailed discussion. Finally, the MLE for dispersion parameter $ q $ can be subsequently obtained from a one-dimensional convex optimization problem, which is computationally tractable.
>
> We wish to further comment that the time complexity result is a major improvement compared to previous choice models aggregated from distance-based ranking models. For example, even when $k=1$, it takes $O(n^2\log n)$ time just to **evaluate** the likelihood under the Mallows model. As for the MLE problem for Mallows, we do not know any method significantly better than the brute force method, whose time complexity is $\Omega (n!)$. That makes the implementation of MLE for Mallows prohibitively difficult. (In sharp contrast, the MLE for the RMJ-based model admits a PTAS for all $k \geq 1$.)
>
>
>
> Q2 (Novelty of the distance function and relationship with Cayley distance)
>
> There are some similarities between RMJ and the Cayley distance. For example, both of them could be represented as a (weighted/unweighted) sum of $(n-1)$ indicator functions. However, RMJ is NOT a weighted version of Cayley: The calculation rules for those indicator functions are different. Under Cayley, $n$ items are classified into certain cycles by their positions, then $(n-1)$ indicator functions are calculated based on cycle comparison. In comparison, under RMJ, the indicator functions are calculated based on the first $(n-1)$ adjacent pairs. For example, the binary vector under Cayley distance of ranking $(3,4,1,2)$ is $(1,1,0)$, while the binary vector under RMJ is $(0,1,0)$. We believe the RMJ vs. Cayley difference is better illustrated by creating an unweighted version of RMJ (referred to as $d_U$ below), and comparing it with Cayley (referred to as $d_C$ below) on concrete examples:
> - $d_C((3,4,1,2)) = 2$ but $d_U((3,4,1,2)) = 1$;
> - $d_C((1,4,3,2)) = 1$ but $d_U((1,4,3,2)) = 2$;
> - $d_C((2,3,4,1)) = 3$ but $d_U((2,3,4,1)) = 1$.
>
> In passing, we wish to mention that the weight itself is part of the innovation. Using the linear decreasing weights, the ranked-choice probabilities, i.e., $P(\pi_k|S)$, have simple closed-form expressions under our RMJ-based ranking model. We believe this theoretical observation is very nontrivial, and to the best of our knowledge, no other distance function has this property.
>
>
>
> Q3 (Linear weight)
>
> We believe that the linear weight makes behavioral sense (e.g., top-positioned deviations carry more weight) and is simple (i.e., being linear). To this end, it makes a nice connection to certain metrics such as DCG. However, the main advantage of linear weight is that it provides a tangible tool to estimate a (mixture of) distance-based ranking model from people's ranked choices from varied display sets. Finally, the RMJ demonstrates favorable performance in capturing and predicting customers' behavior in our case study on real data.

---

> > ### Comment · Reviewer_nVtJ · 2022-08-02
> > **Thanks for the rebuttal**
> >
> > I buy the point that the distance in this paper is different from Cayley. I changed my evaluation (overall score) accordingly.

---

> > > ### Author Response · Authors · 2022-08-03
> > > **Thank you.**
> > >
> > > We thank you for your careful read of the rebuttal and your positive feedback.

---

### Official Review · Reviewer_m77F · 2022-07-15

**Rating:** 6
**Confidence:** 4
**Soundness:** 3 good
**Presentation:** 2 fair
**Contribution:** 3 good

**Summary:**

The authors discuss distance-based ranking model for identifying population's preferences and their ranked choice behavior. This is useful for many applications such as new product introduction, crowdsourcing, etc. They introduce to a new type of distance based (Mallows-type) ranking model, which uses Reverse Major Index (RMJ) as the underlying distance function. They discuss several theoretical properties of this model and show that RMJ based ranking distribution aggregates into (ranked) choice probabilities with simple closed-form expression. In particular, for k > 1, the characterization of choice probabilities, efficient sampling of a top-k list,  and learning of parameters are novel aspects of the paper.

**Questions:**

Overall, the paper has novel aspects. I specially liked Lemma 1 and Lemma 3. However, there a few questions I would like authors to comment on:

Presentation:
1. A lot of things in the introduction are not clear. For example, it is difficult to make sense of the first paragraph on page 2, unless the mallow-type models are not described.
2. Line 28 — what do the authors mean by the most representative distance?
3. What is PTAS in line 77 (it is mentioned in line 227)? Also in line 311, what is MNL? The authors should judicially use acronyms.
4. Use of Mallows-type ranking models and Mallows model for Mallows model with KD distance function is confusing.
5. Line 151 - I do not understand how the authors came to the conclusion that “Therefore, both can be used as “kernels" to smooth out the distribution over rankings.”
6. Theorem 1: Is there a way you can specify that it corresponds to RMJ model. The current form doesn’t tell us that this is for RMJ.

Mathematical Rigorousness:
1. Theorem 2 is not precise. It would be better to add sample complexity numbers. Specifically, quantifying what one means by consistent and infinite samples. I am further confused by the use of the word consistent in Theorem 3. Are the solutions from Mallows smoothing not equivalent? If they are, then the heuristic may not allow to converge to a single entity, but the heuristic may still be consistent when it is converging to equally good set of entities. It would be better to precisely mention (in quantifiable terms) the meaning of these words.

Others:
1. I see that RMJ has some links with measures like NDCG. Also, there can be other variants of RMJ which look at specific portions, or precisely, specific pairwise comparisons of the ranking. For example, pAUC [Narasimhan et al.], pap@k [Hiranandani et al.]. Would authors like to comment on that?
2. Line 147 — Can the authors also mention the use of major index in combinatorics for completeness?

Narasimhan, Harikrishna, and Shivani Agarwal. "A structural SVM based approach for optimizing partial AUC." In International Conference on Machine Learning, pp. 516-524. PMLR, 2013.
Hiranandani, Gaurush, Warut Vijitbenjaronk, Sanmi Koyejo, and Prateek Jain. "Optimization and Analysis of the pAp@ k Metric for Recommender Systems." In International Conference on Machine Learning, pp. 4260-4270. PMLR, 2020.

**Limitations:**

yes.

**Strengths And Weaknesses:**

Strengths:

Introduction of a novel ranking model with certain desired properties
Theoretical and empirical justification of the desired properties
While the presentation may have some issues, but I liked the paper described its results intuitively

Weaknesses:
Presentation: A lot of things are imprecise and not clear in the paper, including introduction
Rigorousness:  Certain mathematical statements are not quantified properly

---

> ### Author Response · Authors · 2022-08-02
> **Thanks and responses from authors!**
>
> We thank you very much for your helpful questions and comments. Please see our responses below.
>
> [Presentation]
>
> Q1 and Q5: The remarks are made in the context of comparing two ways of using ranking models to capture people's choices. The first one, which is quite common, is to identify a **sparse** distribution over rankings to fit the choice data. However, as Antoine et al. (2021) point out, "... Sparse rank-based models cannot account for "noise" or deviations from the $ K $ ranked-lists in the support and tend to over-fit to the observed data. This shortcoming limits their modeling flexibility, resulting in potentially unrealistic predictions." (More discussion is in their paper.)
>
> The second way is "smoothing" the sparse distribution using a distance-based kernel, which is what our approach effectively does. We refer to such models as "Mallows-type," a concept we introduce in line 34. We would also like to mention that the main difference between our method and the one in Antoine et al. (2021) is that while they use the Mallows kernel, we use a kernel based on the reverse major index. We believe it provides a more tangible tool to serve the same purpose (and more, such as modeling ranked choices).
>
>
>
> Q2: While there are many legitimate ranking distances such as Kendall's Tau (KT), Spearman's rank correlation, Spearman's Footrule, etc., KT is arguable the most widely used and extensively studied; see Chierichetti et al. (2018) and references therein. We thank your question and have decided to change the adjective from "representative" to "popular" to more precisely describe our intended meaning.
>
>
> Q3: Thank you for pointing out where we can make the paper clearer. PTAS is short for "Polynomial-Time Approximation Scheme." It is a desirable property for computational complexity: for any fixed error $ \epsilon $, there is a polynomial-time algorithm that achieves $(1-\epsilon)$ optimality of our MLE problem. MNL is short for the Multinomial Logit model, arguably the most popular and natural choice model for benchmarking.
>
>
>
> Q4: Thank you for pointing it out. We notice there is some ambiguity in terminology in the literature: While Mallows (1957) proposed a broader class of ranking models, the term "Mallows model" is sometimes also used to refer to Mallows' $\phi$ model, or the ranking model with KT distance. For clarity, we make the following convention: we use the "Mallows-type model" as the collection of distance-based ranking models (line 34) and the "Mallows model" as the specific model with KT distance.
>
> Q6: Thank you, and we have added the clarification.
>
> [Mathematical Rigorousness]
>
> (Consistency)
> We wish to first clarify that Theorem 2 (as well as Theorem 3) is less about sample complexity but the consistency of the MLE estimator. Here consistency is a statistical concept that means the estimator converges to the ground truth value almost surely (i.e., with probability one) as the sample size converges to infinity. For example, our Theorem 2 has been formally restated in the rebuttal revision.
>
> We believe it is a very intriguing question to pursue sample complexity. While we leave it open for future research, we wish to comment that it is likely to depend heavily on the proportions of display sets since "more informative" display sets help reduce sample complexity. (See Feng et al. (2021) for an example.) In comparison, our results reveal that consistency is only about a certain collection of display sets displayed infinitely many times.
>
> (Equivalence of Mallows smoothing solutions)
> In fact, solutions from Mallows smoothing are NOT equivalent. In the proof Theorem 3 (please see Appendix C for more details), we demonstrate this point using a constructive approach: Suppose the choice data is generated from the Mallows model with the ground truth central ranking $\pi^\ast = (1, 2, ..., n)$. Then even under significant variation of display sets and an infinite amount of choice data, the wrong ranking $\pi^\ast = (1, 2, ..., n-2, n, n-1)$ (i.e., the central ranking with the bottom two items swapped) is a legitimate optimal solution to the Mallows smoothing procedure. This means that there is no tie-breaking rule of Mallows smoothing that guarantees to converge to the ground truth ranking.
>
>
> [Others]
>
> Q1 Yes, these metrics have similarities with RMJ. (NDCG also considers decreasing weight, while pAUC and pAp@k also consider the specific pairwise comparison.) However, it is unclear whether they are "estimatable" from ranked choice data. In particular, it is unknown whether they can aggregate into simple forms of ranked choice probabilities as RMJ.
>
> Q2 A celebrated result for the major index is that it is equidistributed with the number of inversions (i.e., KT distance with the identity ranking). That reveals a deeper connection between RMJ and the KT distance.

---

> > ### Author Response · Authors · 2022-08-09
> > **Thank you.**
> >
> > Thanks again for reviewing our paper. We hope your questions or concerns have been fully addressed. In case you have any additional questions, we are prepared to answer them, too. Thank you very much for your attention.

---

### Meta-Review · Area_Chair_PJAV · 2022-08-23

**Recommendation:** Accept
**Confidence:** Certain

**Metareview:**

This paper makes a contribution to probabilistic models for ranking data.  The authors propose a new distribution similar to the  Mallows model but with the so-called reverse major index instead of Kendall as distance function. They address the problem of ML estimation for ranking and choice and show formal consistency properties of the estimate. Simulation studies are also included. The paper has been well-received by all authors. Although a few critical comments have been raised in the original reviews, these could be dispelled in the rebuttal phase.

That said, a critical discussion came up in the final phase of decision making (regrettably too late to enquire the authors on this point).  Here, it was noticed that the new distance function proposed by the authors seems to have rather doubtful properties. In particular, neither symmetry nor the triangle inequality hold. For example The distance between (1,2,...,n-1,n) and (2,3,,...,n,1) is 1, but the distance between  (2,3,...,n,1) and (1,2,...,n) is n-1. Somewhat counterintuitive behaviour of the distance can also be observed in other cases. For example, swapping the last two items has the same effect as moving the top-item to the bottom: d (1, 2, ..., n, n-1) = d( 2, 3, ..., n, 1) = 1. Or, swapping the second and third item leads to a much higher distance than moving the top-item to the bottom: d(2, 3, ..., n, 1) = 1 < d(1, 3, 2, 4, ..., n) = n-2. Maybe such properties could be explained or defended, but they should at least be addressed in the paper. Comparing Kendall and the new distance, the authors write: "It is difficult to tell which kernel is “better" from an axiomatic approach as both distance functions satisfy the basic axioms for ranking distances". This is at least highly misleading, to put it mildly. Note that the axiomatic approach to ranking distances goes back to Kemeny, who required quite a number of properties and showed that the Kendall distance is the UNIQUE distance satisfying all properties. So comparing Kendall with a distance that violates almost all of these properties, it should be easy to say which one is better from an axiomatic perspective ...

**Award:**

No

---

### Decision · Program_Chairs · 2022-09-14

Accept